# The giant diploid faba genome unlocks variation in a global protein crop

Murukarthick Jayakodi[1,26], Agnieszka A. Golicz[2,26], Jonathan Kreplak[3,26], Lavinia I. Fechete[4], Deepti Angra[5], Petr Bednář[6], Elesandro Bornhofen[7], Hailin Zhang[1], Raphaël Boussageon[3], Sukhjiwan Kaur[8], Kwok Cheung[5], Jana Čížková[9], Heidrun Gundlach[10], Asis Hallab[11,12], Baptiste Imbert[3], Gabriel Keeble-Gagnère[8], Andrea Koblížková[13], Lucie Kobrlová[14], Petra Krejčí[6], Troels W. Mouritzen[4], Pavel Neumann[13], Marcin Nadzieja[4], Linda Kærgaard Nielsen[15], Petr Novák[13], Jihad Orabi[16], Sudharsan Padmarasu[1], Tom Robertson-Shersby-Harvie[5], Laura Ávila Robledillo[13], Andrea Schiemann[16], Jaakko Tanskanen[17], Petri Törönen[18], Ahmed O. Warsame[5], Alexander H. J. Wittenberg[19], Axel Himmelbach[1], Grégoire Aubert[3], Pierre-Emmanuel Courty[3], Jaroslav Doležel[9], Liisa U. Holm[18], Luc L. Janss[7], Hamid Khazaei[17], Jiří Macas[13], Martin Mascher[1,20], Petr Smýkal[14], Rod J. Snowdon[2], Nils Stein[1,21], Frederick L. Stoddard[22,23], Jens Stougaard[4], Nadim Tayeh[3], Ana M. Torres[24], Björn Usadel[11,25], Ingo Schubert[1], Donal Martin O'Sullivan[5✉], Alan H. Schulman[17,18,23✉] & Stig Uggerhøj Andersen[4✉]

Increasing the proportion of locally produced plant protein in currently meat-rich diets could substantially reduce greenhouse gas emissions and loss of biodiversity[1]. However, plant protein production is hampered by the lack of a cool-season legume equivalent to soybean in agronomic value[2]. Faba bean (*Vicia faba* L.) has a high yield potential and is well suited for cultivation in temperate regions, but genomic resources are scarce. Here, we report a high-quality chromosome-scale assembly of the faba bean genome and show that it has expanded to a massive 13 Gb in size through an imbalance between the rates of amplification and elimination of retrotransposons and satellite repeats. Genes and recombination events are evenly dispersed across chromosomes and the gene space is remarkably compact considering the genome size, although with substantial copy number variation driven by tandem duplication. Demonstrating practical application of the genome sequence, we develop a targeted genotyping assay and use high-resolution genome-wide association analysis to dissect the genetic basis of seed size and hilum colour. The resources presented constitute a genomics-based breeding platform for faba bean, enabling breeders and geneticists to accelerate the improvement of sustainable protein production across the Mediterranean, subtropical and northern temperate agroecological zones.

Faba bean (*Vicia faba* L., 2*n* = 12) was domesticated in the near East more than 10,000 years BP[3,4] and its broad adaptability, value as a restorative crop in rotations and high nutritional density have propelled it to the status of a global crop grown on all continents except Antarctica[5]. Despite its global importance, no extant wild progenitor has been found. Nonetheless, the finding of Neolithic charred wild faba bean seeds points to pre-domestication use of this species by hunter–gatherers and possible domestication in the Levant[3]. The presence of several closely related species (*Vicia narbonensis*, *Vicia palaestina* and *Vicia kalakhensis*) in the same region[6] gives hope that a wild progenitor may yet be found. Faba bean exhibits such extreme variation in seed size that some taxonomists defined the primitive, small-seeded 'paucijuga' forms[7] or small-seeded 'minor' forms[8] as separate subspecies from the medium–large 'faba' types. However, the absence of reproductive barriers between any of these forms means that 'major', 'minor', 'equina' and 'paucijuga' forms are now regarded as botanical types resulting from sustained human selection on growth habit and seed size over many thousands of years[9]. Faba bean continues to be relevant in the twenty-first century as humanity strives to lower agricultural greenhouse gas emissions by replacing meat or milk protein with plant-based alternatives[10]. It is the highest yielding of all grain legumes[11] and has a favourable protein content (approximately 29%) compared with other cool-season pulses such as pea, lentil and chickpea, making it a suitable candidate to meet challenging projected future protein demands. Furthermore, the high biological nitrogen fixation rates of faba bean[12] and the long duration of nectar-rich, pollinator-friendly flowers[13] provide important ecosystem services, which means that cultivation of faba bean is increasingly seen as key for sustainable intensification strategies. Conversely, its partially allogamous mating system and estimated 13-Gb genome size, coupled with a low seed multiplication

rate, have made it a challenging target for breeders[14]. Substantial progress has been made in faba bean genomics and pre-breeding research. The mining of the first faba bean transcriptomes and development of single-nucleotide polymorphism (SNP)-based genetic maps, which showed strong collinearity with model legumes, set the scene for the identification of the WD40 transcription factor underlying the *Zero Tannin1* locus[15], whereas a combination of high-resolution mapping, transcriptomic and metabolomic approaches led to the cloning of the *VC1* gene, which controls seed content of the antinutrients vicine and convicine, paving the way for safer exploitation of the crop in the human food chain[16]. However, the lack of a reference genome sequence greatly complicated these studies, and improved faba bean genomic resources are urgently needed to accelerate crop improvement.

## Sequence of the giant faba bean genome

The 13-Gb faba bean genome ($2n = 2x = 12$) is one of the largest diploid field crops (Extended Data Fig. 1a,b) and its dominant repeat family members are longer[17,18] (up to 25 kb) than those in similarly sized polyploid cereal genomes[19]. The biggest of its six chromosomes holds the equivalent of an entire human genome. Although aiding cytogenetics[20], these properties made genome assembly very challenging before the emergence of long and accurate reads. We chose the inbred line 'Hedin/2' as a reference genotype owing to its high autofertility and productivity, combined with an early maturing spring habit and exceptional degree of homozygosity. We sequenced its genome with PacBio HiFi long reads to 20-fold coverage and assembled 11.9 Gb of sequence, more than half of which was represented by contigs longer than 2.7 Mb (Extended Data Table 1). Linkage information afforded by a genetic map (Supplementary Table 1) and chromosome conformation capture sequencing (Hi-C) data placed 11.2 Gb (94%) into chromosomal pseudomolecules (Fig. 1a and Supplementary Fig. 1a). Chromatin immunoprecipitation sequencing for centromeric histone H3 pinpointed the locations of the centromeres in the Hedin/2 assembly, and arm ratios were consistent with karyotypes (Supplementary Fig. 1b). The single metacentric chromosome 1 was the only one to adopt a Rabl configuration, evident from the presence of both a main and an anti-diagonal on that chromosome in Hi-C interaction plots (Fig. 1a). This supports the notion that chromosome arms need to be of approximately equal size to spatially juxtapose in interphase. Some regions of the Hi-C contact matrices were empty for lack of mapped short reads (Fig. 1a). These white regions coincided with the locations of enormous (up to 752 Mb) satellite arrays and aligned well with cytological maps of those repeats (Fig. 1b,c). Assembly evaluation with Merqury[21] revealed the genome to be 96.3% complete, with a consensus quality value of 60.5, indicating a high accuracy of our Hedin/2 assembly (Extended Data Table 1 and Extended Data Fig. 1c). A good collinear agreement between the genetic and physical maps further validated the accurate assignment of contigs to chromosomes (Supplementary Fig. 1a). In addition, the long terminal repeat (LTR) assembly index score of 10.5 supports the contiguity of our assembly. We also collected HiFi data (tenfold coverage) for the German cultivar 'Tiffany' and assembled these into a set of contigs with an N50 of 1.6 Mb, spanning 11.4 Gb (Extended Data Table 1). Similar to Hedin/2, Merqury assessments supported the high quality of the Tiffany assembly (Extended Data Table 1 and Extended Data Fig. 1d). This level of completeness and contiguity was sufficient to arrange the contigs into pseudomolecules guided by the Hedin/2 reference (Extended Data Table 1 and Extended Data Fig. 2). In the future, the Hedin/2 assembly is expected to become the nucleus of a faba pan-genome.

## Driving forces of genome size expansion

The genome sequence of Hedin/2 was annotated with RNA sequencing data from nine diverse tissues (Supplementary Table 2), resulting in a total of 34,221 protein-coding genes (Supplementary Table 3).

A similar number of gene models (34,043) was also predicted in the Tiffany assembly. The predicted Hedin/2 gene models captured 96% of single-copy orthologues conserved in Embryophyta according to the BUSCO metric (Supplementary Table 4). Gene density was uniform along the chromosomes (except for the positions of satellite DNA arrays) without the proximal–distal gradient typically observed for grass chromosomes[22]. Meiotic recombination displayed a similar distribution with an average of 27 genes per centimorgan (Fig. 1d and Extended Data Fig. 3). Thus, despite its large genome, faba bean may be more amenable to genetic mapping than cereals, in which up to one-third of genes are locked in non-recombining pericentric regions[22]. Gene order was highly collinear and syntenic with other legumes (Fig. 2a). To further validate gene annotation, we aligned 262 *Medicago truncatula* genes related to symbiosis with rhizobia or arbuscular mycorrhizal fungi and found putative orthologues for them all. In addition, using RNA sequencing, we verified that a large subset of these genes was responsive to inoculation, as expected[23–25] (Supplementary Table 5).

In contrast to gymnosperms, with similarly gigantic genomes[26,27], introns in faba bean genes were not larger than in angiosperms with smaller genomes (Fig. 2b), but the intergenic space was more expanded (Fig. 2c). Moreover, the number of multicopy gene families in faba bean was similar to related diploid species (Supplementary Table 6 and Supplementary Fig. 2), in contrast to soybean, which is considered a partially diploidized tetraploid[28]. Likewise, nucleotide substitution rates between paralogous and orthologous gene pairs place the last whole-genome duplication (WGD) event in the faba bean lineage at 55 million years ago (Ma), well before the split from other Papilionoideae[29] (Fig. 2e and Supplementary Fig. 3), a taxon that also includes pea and lentil (*Lens culinaris*), species from which faba bean diverged around 12.2 and 13.8 Ma, respectively. Although we did not find evidence for a recent WGD in faba bean, more genes were duplicated in tandem than in pea and lentil (Supplementary Fig. 4a). These duplications post-date the last WGD and occurred later than tandem duplications in *Arabidopsis thaliana* and *M. truncatula* (Supplementary Fig. 4b), two species whose genomes were also rich in such events and coincided with recent transposable element (TE) expansion. Overall, there were 1,108 syntenic clusters of tandemly duplicated genes in Hedin/2 and Tiffany, some of which differed in copy number. Of note, the agronomically relevant family of leghaemoglobins had expanded (Supplementary Table 7). Despite this, the absence of a lineage-specific WGD or widespread gene family expansion means that the proliferation of repeat elements largely explains why the faba genome is more than seven times larger than that of its close relative common vetch (*Vicia sativa*)[30].

Approximately 79% of the Hedin/2 assembly was annotated as transposon-derived (Supplementary Table 8). By far, the largest group is the LTR retrotransposons (RLX), accounting for 63.7% of the genome sequence. Other groups of TEs represent only minor fractions of the genome (Supplementary Table 8). Among the RLX, those of the *Gypsy* (RLG) superfamily outnumber *Copia* (RLC) elements by more than 2:1 (Fig. 1d and Extended Data Fig. 3). The *Ogre* family of *Gypsy* elements alone make up almost half (44%) of the genome, confirming its status as a major determinant of genome size in the Fabaceae[18] (Fig. 2f). The great length of individual elements (up to 35 kb for *Ogre* and 32 kb of *SIRE*, the longest and second-longest elements, respectively), together with their abundance, partially explains the large size of the faba bean genome (Supplementary Fig. 5). In addition, a large and diverse set of satellite repeat families that differ in their monomer sequences and genome abundance[31] accounted for 9.4% of the total assembly length, with the most abundant satellite family *FokI* representing 4% (0.475 Gb). *FokI*, together with several other highly amplified satellites, forms prominent heterochromatic bands on faba bean chromosomes (Fig. 1c). The TE density was remarkably invariable along all six chromosomes, mirroring gene density and recombination rate, and inverse to the density of satellite arrays (Fig. 1d and Extended Data Fig. 3).

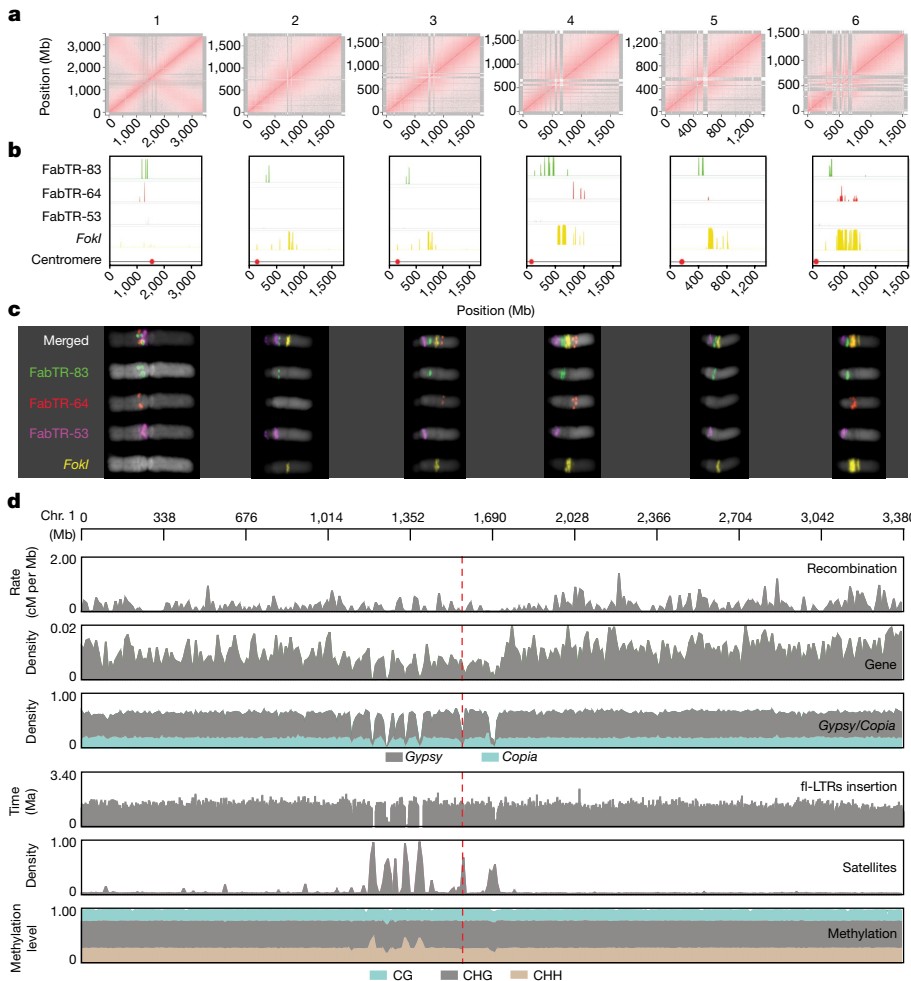

**Fig. 1 | Gigabase-size chromosome-scale assembly of faba bean.**
**a**, Intrachromosomal contact matrix of assembled chromosomes. The red colour intensity indicates the normalized Omni-C Hi-C links between 1-Mb windows on each chromosome. The antidiagonal pattern in chromosome 1 represents the Rabl configuration. **b**, Distribution of major families of satellite repeats (FabTR-83 in green, FabTR-64 in red, FabTR-53 in magenta and *FokI* in yellow).

**c**, Distribution of major families of satellite repeats on metaphase chromosomes visualized by multicolour fluorescent in situ hybridization. **d**, Distribution of genomic components including recombination (cM per Mb), gene density, LTR retrotransposons of *Gypsy* and *Copia*, full-length LTR-retrotransposon (fl-LTR) insertions, satellite repeats and DNA methylation (CH, CHG and CHH context) on chromosome 1. The red dashed line represents the centromere position.

The persistence of retrotransposons as full-length copies can tell us about the balance between genome size expansion by retrotransposition and shrinkage by elimination through recombination. Modelling the solo-LTRs (sLTRs) as the product of the recombination between the LTRs of a single element, and assuming the canonical *Ogre* to comprise LTRs of 4,161 bp and an internal domain of 11,655 bp, the 395,657 sLTRs represent a loss of 6.26 Gb of DNA from the genome (55.6% of the current assembly size). This loss would be even greater if recombination between LTRs of different individual *Ogre* elements as well as DNA double-strand break repair-mediated internal truncations were considered. However, unlike plant species with smaller genomes, there were generally many fewer sLTRs in faba bean relative to the number of full-length LTRs, similar to large gymnosperm genomes (Fig. 2g), indicating slower removal than spreading of RLX[32]. The *V. sativa* genome of 1.65 Gb was earlier reported to comprise 22.5% of *Ogre* elements and to have 1.6 sLTRs for each full-length *Ogre*, considerably more than found for *V. faba*[18].

## Efficient genome-wide methylation

In addition to the relatively slow RLX elimination rate, it is also possible that lower levels of methylation could have accelerated TE proliferation through less efficient silencing. We found that most cytosines in the faba bean genomes were methylated: 95.8% in CG, 88.2% in CHG and 14% in CHH contexts (Fig. 1d and Extended Data Fig. 3), placing it among the most highly methylated plant genomes[32]. Gene body methylation followed the canonical pattern (Fig. 3a) observed in other plants[33]: CG methylation was enriched in internal exons and introns (Supplementary Fig. 6a), in contrast to low methylation in first exons, and may be related to transcriptional repression[34]. Genes with a high level of gene body methylation were more highly expressed in young leaf tissue (Supplementary Fig. 6b) and also tended to be longer (average 3.3 kb). The elements of the major superfamilies of RLX, *Gypsy* and *Copia*, occupied 48% and 11% of the genome, respectively. They were also heavily methylated, more so in their bodies than in their flanking regions (Fig. 3b). The most recent transposon burst occurred less than 1 Ma, but many structurally intact elements were between 3 and 5 million years old (Fig. 3c). Both young and old insertions were invariably methylated in all three sequence contexts (Extended Data Fig. 4a). In contrast to other plant taxa[35], RLX insertion times and methylation levels were uncoupled. Conspicuous islands of elevated CHH methylation also coincided with the abundant satellite repeat FabTR-83 (Extended Data Fig. 4b), which accounts for 1.1% of the genome. Generally, the faba bean methylation machinery

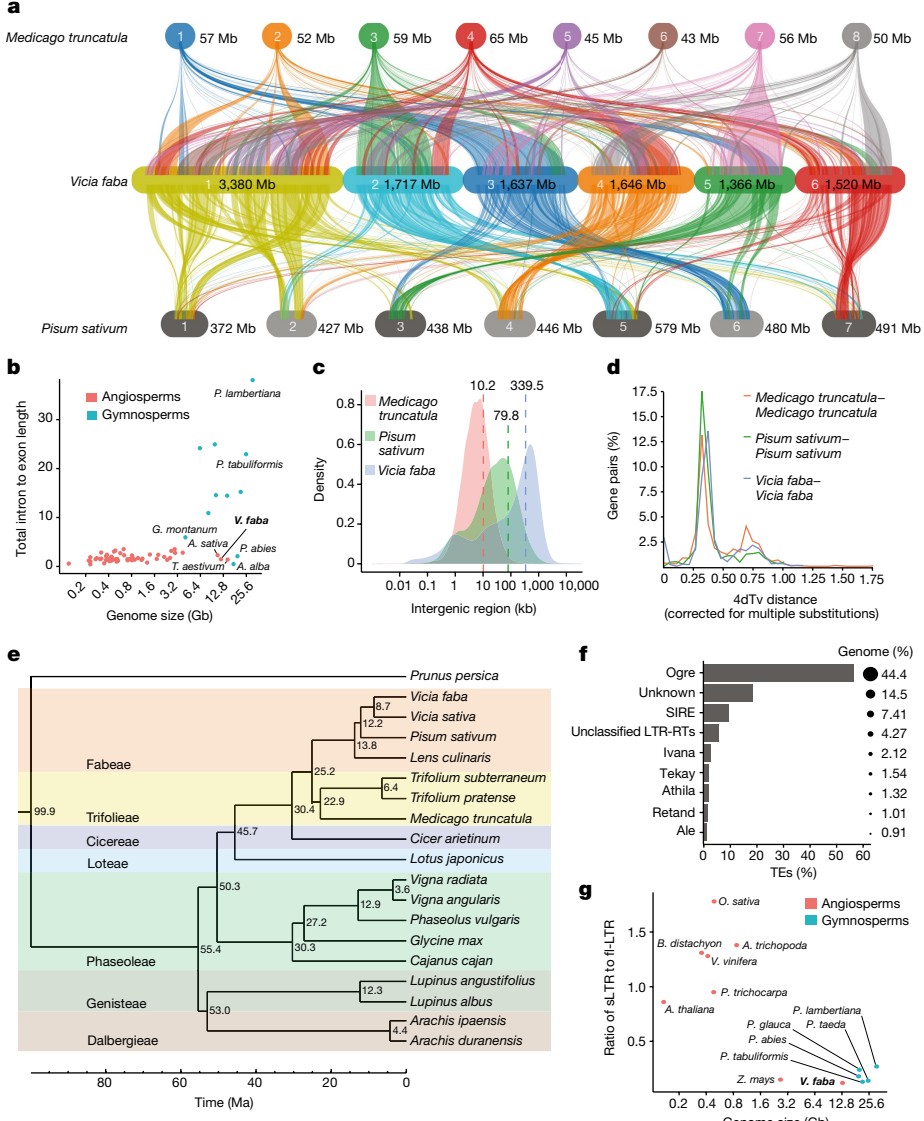

**Fig. 2 | Evolution and synteny analysis in faba bean. a**, Syntenic relationship of faba bean (middle) with *Medicago* (top) and pea (bottom). **b**, Intron versus exon lengths across angiosperms and gymnosperms. *A. alba*, *Abies Alba*; *A. sativa*, *Avena sativa*; *G. montanum*, *Geum montanum*; *P. abies*, *Picea abies*; *P. lambertiana*, *Pinus lambertiana*; *P. tabuliformis*, *Pinus tabuliformis*; *T. aestivum*, *Triticum aestivum*. **c**, Lengths of legume intergenic regions. **d**, Distribution of the transversion rates at the fourfold degenerate sites (4dTv) of paralogous gene pairs. **e**, Phylogenetic relationships between faba bean and other crop legumes in the Papilionoideae clade. The numbers on the branches indicate the estimated divergence time (Ma). **f**, Summary of the faba bean retrotransposon composition by family. **g**, Ratio of sLTR to fl-LTR plotted against genome size in gymnosperms and angiosperms. The ratio for other species was retrieved from ref. [32]. *A. trichopoda*, *Amborella trichopoda*; *B. distachyon*, *Brachypodium distachyon*; *O. sativa*, *Oryza sativa*; *P. abies*, *Picea abies*; *P. glauca*, *Picea glauca*; *P. taeda*, *Pinus taeda*; *P. trichocarpa*, *Populus trichocarpa*; *V. vinifera*, *Vitis vinifera*; *Z. mays*, *Zea mays*.

appeared fully functional, efficiently methylating all classes of repetitive elements, suggesting that methylation deficiency is unlikely to have a role in genome expansion. This is supported by investigation of genes involved in RNA-directed DNA methylation[36], for which gene copy number in faba bean is similar to *V. sativa*, pea and lentil (Supplementary Table 9).

## Integration of QTL and variation data

The faba bean genome sequence provides a unified frame of reference for genetic mapping, gene expression profiling and comparative genomics. To assist the adoption of the new infrastructure among faba bean breeders and geneticists, we mapped markers from two commonly used genotyping platforms, the Illumina Infinium 1,536 SNP and the Illumina Oligo Pool Array assays. Moreover, we projected genetic maps

of both different biparental crosses and derived consensus genetic maps onto the genome assembly. This provided physical coordinates to quantitative trait loci (QTLs) for disease resistance and phenology. Marker maps and QTL intervals can be browsed interactively at https://pulses.plantinformatics.io (Supplementary Fig. 7 and Supplementary Note 1).

The genome sequence has also paved the way for sequence-based genotyping. We mined the Hedin/2 assembly for oligonucleotide probes for use in single primer enrichment technology (SPET)[37], a reduced-representation genotyping method with high throughput and low per-sample costs. A panel of 197 cultivated accessions from a diversity collection designed for trait mapping was profiled with a 90,000 probe SPET assay with at least one probe in each predicted gene (Supplementary Table 10). Sequence reads were mapped to the Hedin/2 assembly and 1,081,031 segregating variants (SNPs) uniformly

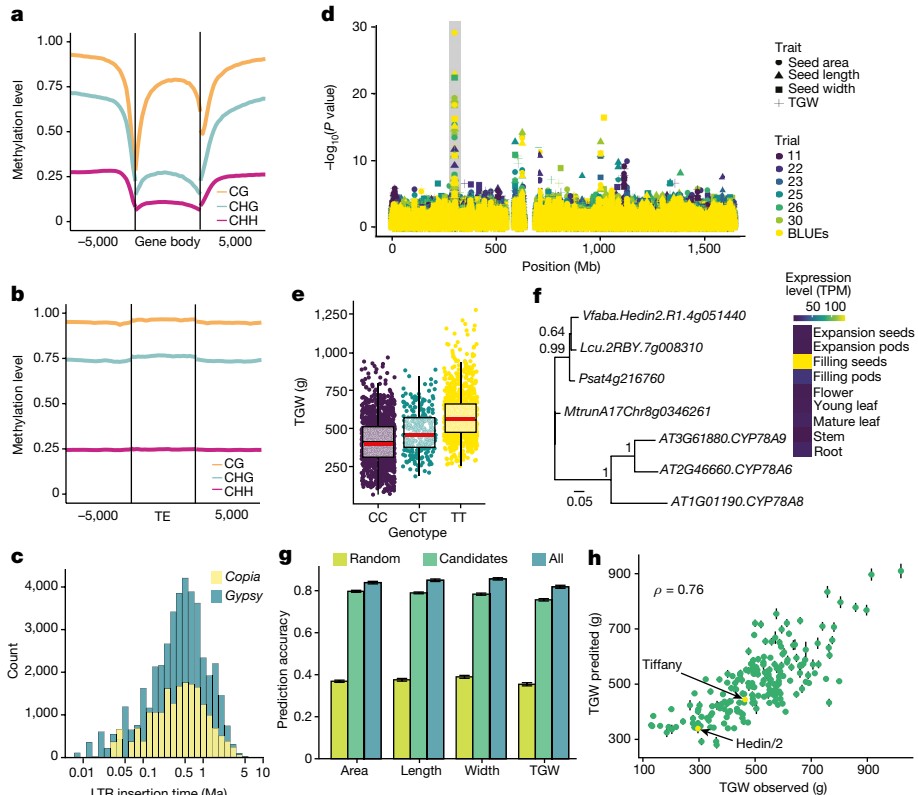

**Fig. 3 | TE methylation and seed size genetics. a**, Global distribution of DNA methylation levels at protein-coding genes including a 5-kb region upstream of the transcription start site and downstream of the transcription end site. **b**, DNA methylation patterns for TEs and their 5-kb flanking regions. **c**, Distribution of *Copia* (RLC) and *Gypsy* (RLG) retrotransposons based on insertion time. **d**, Combined Manhattan plot of GWAS analysis of seed area, seed width, seed length and thousand grain weight (TGW) for chromosome 4. BLUE, best linear unbiased estimator. **e**, Effect plot for TGW for the SNP marker within *Vfaba.Hedin2.R1.4g051440* at position 299,823,118 on chromosome 4 (highlighted by a grey bar in part **d**). $n$ = 2,499 data points distributed over six trials. In the box plots, the horizontal black central line represents the median, the red line indicates the mean, the box ranges from the first to third quartile, and the vertical black lines extend to the smallest or largest point within the

1.5× interquartile range. **f**, Phylogenetic tree showing the relationships between *Vfaba.Hedin2.R1.4g051440* and its orthologues in pea, lentil, *Medicago* and *Arabidopsis* and its expression levels in nine diverse tissues of Hedin/2 in transcripts per million (TPM). The branch lengths are measured in the number of substitutions per site. Numbers next to branches indicate bootstrapping support. **g**, Genomic best linear unbiased prediction accuracies for seed traits using 15 randomly chosen SNPs (random), the 15 seed-size-associated markers (candidates) or all markers (all). **h**, Mean genomic best linear unbiased prediction of TGW using the 15 seed-size-associated SNPs plotted against the mean observed values. The Pearson correlation coefficient is indicated. Error bars indicate the standard deviation of a fivefold genomic best linear unbiased prediction cross-validation scheme with predicted values computed from ten replicate runs (**g**,**h**; see Methods).

distributed along the genome were called. Analysing the functional impact of SNPs and short insertions and deletions (indels) found in genic regions, we identified 1,042 SNPs and 65 indels introducing a premature termination codon in at least one of 197 accessions. The premature termination codons interrupted transcripts of 933 genes, including 39 resistance gene analogues (Extended Data Fig. 5 and Supplementary Table 11). We provide a full atlas of genes and accessions carrying premature termination codons to facilitate functional studies (Supplementary Table 12).

## Genetics of seed size

Despite impressive variation and critical agronomic importance, the genetics underlying faba bean seed size have remained obscure, with only a few large seed weight QTL regions detected[38]. We collected seed size data for the 197 accessions at two locations for 3 years and combined these with the SPET marker data to carry out a high-resolution genome-wide association study (GWAS). This identified 15 marker–trait associations, which were stable across trials and GWAS methods (Fig. 3d, Extended Data Fig. 6, Supplementary Figs. 8–11 and Supplementary Tables 13 and 14). The most prominent signal was located on chromosome 4 within the *Vfaba.Hedin2.R1.4g051440* gene (Fig. 3d,e

and Supplementary Fig. 12), which is highly expressed in faba bean seeds, resides within a previously identified seed weight QTL region[38] (Supplementary Note 1) and is homologous to the *Arabidopsis CYP78A* genes known to regulate seed size[39] (Fig. 3f). *Vfaba.Hedin2.R1.4g051440* is thus likely to contribute to seed size variation in faba bean, but does not explain the majority of the variation for this complex trait (Fig. 3e). By contrast, using all 15 high-confidence markers, we were able to predict seed size with nearly as high accuracy as when using the full set of genomic markers (Fig. 3g,h), indicating that we have identified a large proportion of the key loci, and associated candidate genes, controlling faba bean seed size. To investigate whether seed size has been a driver of population differentiation, we carried out population structure analysis by model-based ancestry estimation, and principal component analysis divided the diversity panel into four groups, corresponding to their geographical origin (Extended Data Fig. 7). All populations had a similar proportion of seed-enlarging alleles and all seed-enlarging alleles were present in all populations, with the exception of population 4, which comprised relatively few accessions that all harboured the seed-enlarging allele of *Vfaba.Hedin2.R1.4g051440* (Extended Data Fig. 8 and Supplementary Table 13). This allele distribution across populations suggests extensive historical sharing of germplasm by breeders across geographical regions.

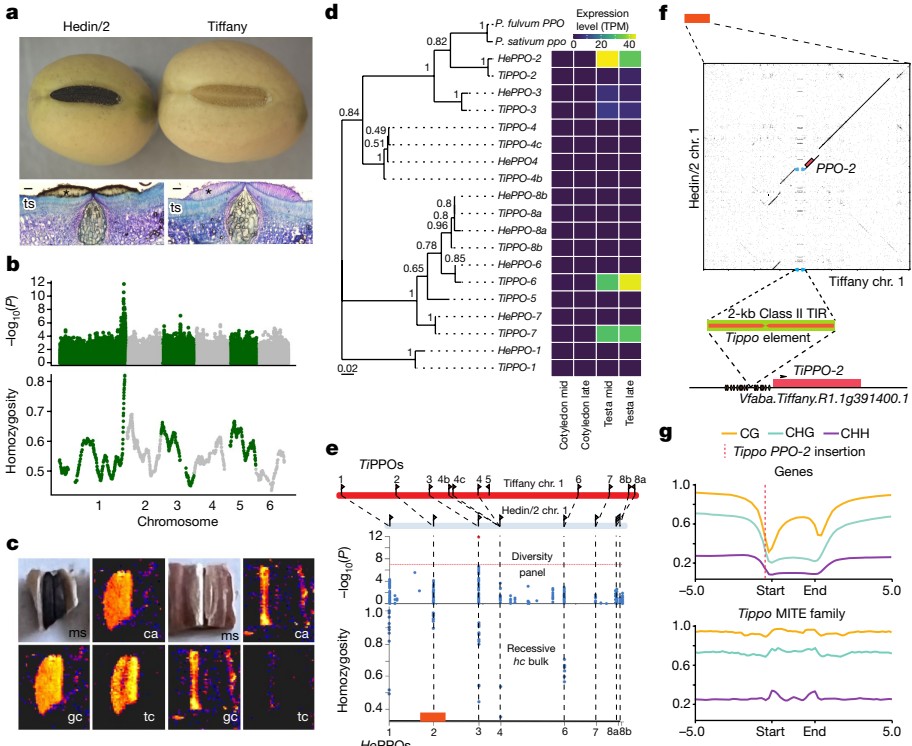

**Fig. 4 | Rearrangements at the complex *PPO* locus give rise to changes in *PPO* gene expression and hilum colour. a**, Whole seeds of dark hilum Hedin/2 and pale hilum Tiffany are shown above light microscope images of a transverse section (ts) of the dark (left) and pale (right) hila. Asterisks indicate the counter palisade cells of the hilum where PPO activity is indicated by brown pigmentation. Scale bars, 20 μm. **b**, GWAS with hilum colour scored as a binary trait in the NORFAB diversity panel (top) and for homozygosity of pale hilum parent alleles in an 84-component recessive pseudo-$F_2$ pale hilum bulk (bottom). **c**, Optical image of Hedin/2 (left) and Tiffany (right) hilum specimens subjected to laser desorption–ionization mass spectrometry imaging (ms), and the laser desorption–ionization mass spectrometry imaging signal distribution for chlorogenic acid (ca), epi-gallocatechin (gc) and tetracosylcaffeate (tc), showing absence of signal from these compounds from the hilum area of the pale hilum genotype. **d**, Phylogenetic tree showing the relationships between the causative pea gene and 8 and 11 PPO copies found in a tandem arrangement

at the Hedin/2 and Tiffany *HC* locus, respectively. *He* and *Ti* prefixes denote Hedin/2- and Tiffany-specific versions of *PPO* paralogues. The branch lengths are measured in the number of substitutions per site. Numbers next to branches indicate bootstrapping support. **e**, From top to bottom: to-scale schematic of the chromosome 1 PPO cluster showing the order and orientation of PPOs in Tiffany and Hedin/2, with syntenic PPO copies joined by dashed lines; close-up of hilum colour associations in the NORFAB diversity panel and homozygosity in the pale hilum (*hc*) bulk are also shown. The red block shows the region expanded in the dot plot in panel f. **f**, Dot plot of 20 kb upstream and downstream of *HePPO-2* (3,291,947,464) and *TiPPO-2* (3,263,562,398) showing an approximately 2-kb MITE, named '*Tippo*', inserted in Tiffany among predicted transcription factor-binding sites (brown ovals) in close proximity to the RNA polymerase II-binding site (TATA box is shown as a green oval) and transcription start site (arrowhead) of *PPO-2*. TIR, Terminal inverted repeat. **g**, Genome-wide methylation status of genes (top) compared with the *Tippo* MITE family (bottom).

## Hilum colour candidate-gene mapping

The two sequenced genotypes, Hedin/2 and Tiffany, differ not only in seed size but also in seed hilum colour (Figs. 3h and 4a). This is an important Mendelian quality trait, with pale hila being preferred by human consumers[40]. Similarly to seed size, no candidate genes have yet been identified. To reveal the molecular basis of the trait, we carried out a GWAS for hilum colour and identified a single prominent peak that was coincident both with the previously mapped *Hilum Colour (HC)* locus[40] and peak homozygosity in a recessive pale hilum bulk of segregants from a cross between pale and dark hilum faba bean varieties (Fig. 4b). We found the most highly associated GWAS marker in a polyphenol oxidase (*PPO*) gene residing in a cluster of eight fully intact and highly conserved *PPO* genes in the Hedin/2 assembly. In pea, PPO variation controls hilum colour. At the syntenic *PI* locus, a frameshifted, non-functional form of the single *PPO* copy conferring a pale hilum is fixed in all modern pea varieties[41]. The pattern of pigmentation (Fig. 4a) and content of oligomeric phenolic compounds such as dimers and trimers of chlorogenic acid, gallocatechin and tetracosylcaffeate on the surface of pigmented and non-pigmented hila in faba bean (Fig. 4c and Extended Data Fig. 9) were very similar to those observed in pea[40]. Together with the genetic data, this indicates

that differential PPO activity is responsible for hilum colour variation in both pea and faba bean, but it was unclear which faba bean PPO (or PPOs) may be causative.

To clarify, we compared the phylogeny and structure of the PPO clusters of the two fully sequenced genotypes Hedin/2 (dark) and Tiffany (pale). *VfPPO-2* shared the highest level of identity with the causal pea gene *Psat1g2063360* (Fig. 4d), whereas the most strongly associated GWAS marker was found in *VfPPO-3*, and the pale hilum bulk homozygosity peak was localized between *VfPPO-2* and *VfPPO-3*, suggesting that the causal polymorphism resided at the proximal end of the cluster (Fig. 4e). Structurally, apart from large differences in intergenic distances between syntenic PPO genes caused mainly by *Ogre* insertions, the most striking features of the Hedin/2–Tiffany comparison were the triplication of *VfPPO-4* in Tiffany and the absence of *VfPPO-5* in Hedin/2 (Fig. 4e), prompting us to investigate whether these structural variations were associated with variation in PPO gene expression. We first established that transcription of the PPO gene cluster was almost exclusively confined to the maternal testa tissue (which encompasses the hilum), rather than the cotyledon in both genotypes (Fig. 4d, Supplementary Tables 15 and 16 and Supplementary Fig. 13). In Hedin/2 testa, *VfPPO-2*, and to a lesser extent *VfPPO-3*, accounted for nearly all PPO expression. By contrast, Tiffany testa PPO expression

was dominated by *VfPPO-6* and *VfPPO-7* (Fig. 4d). A detailed annotation and comparative repeat analysis of the PPO cluster region (Supplementary Fig. 14) highlighted an approximately 2-kb AT-rich MITE insertion in the *TiPPO-2* promoter region (Fig. 4f), which interrupts the sequence of the predicted *VfPPO-2* promoter and belongs to a class of MITE associated with high levels of methylation (Fig. 4g). Together, our results suggest that regulation of expression of *VfPPO-2* controls hilum colour variation in faba bean. Beyond suggesting a causative mechanism for pale hilum in faba bean, our analysis illustrates that increased copy number does not necessarily correlate with trait expression and emphasizes the utility of complete genome sequences from multiple genotypes.

## Discussion

Faba bean is one of the earliest domesticated crops. It was part of the Neolithic package of crops that the early farmers took with them as they left the fertile crescent[42]. Concern about faba bean toxicity was voiced in classical antiquity[43]. In the twenty-first century, nutritional quality remains a central breeding goal: new faba bean varieties should be low in the alkaloid glycosides vicine and convicine as well as in tannins. Furthermore, essential amino acids should be balanced better to accommodate human dietary needs, whereas seed phytate and protease inhibitors should be reduced to improve nutrient bioavailability, all the while taking care not to alter seed size or compromise pest resistance and at the same time improving yield stability. Faba bean breeders can now face these complex challenges enabled by genomic resources and insights. Ubiquitous and frequent recombination will allow rapid introgression of new traits into elite material and permits powerful and broadly applicable mapping approaches exploiting the high SNP densities provided by SPET genotyping. Pinpointing causative variants can still be difficult in genomic regions with tandemly duplicated genes, but our investigation of hilum colour demonstrates that these challenges can be overcome using high-quality long-read assemblies coupled with transcriptomics. Repeats and their methylation influence genome evolution, but can also affect gene expression variation where the repeat elements insert within the regulatory regions of genes. Our rich genome-wide repeat annotation now sheds light on these effects, adding an important component to the genomics-based breeding platform. Faba bean appears to be an isolated species and does not hybridize with others in the genus *Vicia*[38], effectively barring the use of wild relatives in faba bean breeding. However, stable *Agrobacterium*-mediated transformation of faba bean embryo axes has been reported[44]. Together with target gene identification, this opens up the possibility of gene editing. Expanding the platform further by cataloguing and exploiting as much of the segregating variation of domesticated faba bean as possible is especially important, as we do not know its wild progenitor. Population-scale resequencing of mutants, genebank collections and elite cultivars, along with pan-genome assemblies of representatives of major germplasm groups, can now proceed, supported by the resources and methods presented here.

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

[1]Leibniz Institute of Plant Genetics and Crop Plant Research (IPK) Gatersleben, Seeland, Germany. [2]Department of Plant Breeding, Justus Liebig University Giessen, Giessen, Germany. [3]Agroécologie, INRAE, Institut Agro, University Bourgogne, University Bourgogne Franche-Comté, Dijon, France. [4]Department of Molecular Biology and Genetics, Aarhus University, Aarhus C, Denmark. [5]School of Agriculture, Policy and Development, University of Reading, Reading, UK. [6]Department of Analytical Chemistry, Faculty of Science, Palacky University, Olomouc, Czech Republic. [7]Center for Quantitative Genetics and Genomics, Aarhus University, Aarhus C, Denmark. [8]Agriculture Victoria, AgriBio, Centre for AgriBioscience, Bundoora, Victoria, Australia. [9]Institute of Experimental Botany of the Czech Academy of Sciences, Centre of the Region Haná for Biotechnological and Agricultural Research, Olomouc, Czech Republic. [10]Plant Genome and Systems Biology (PGSB), Helmholtz Center Munich, German Research Center for Environmental Health, Neuherberg, Germany. [11]IBG-4 Bioinformatics Forschungszentrum Jülich, Jülich, Germany. [12]Bingen Technical University of Applied Sciences, Bingen, Germany. [13]Biology Centre, Czech Academy of Sciences, Institute of Plant Molecular Biology, České Budějovice, Czech Republic. [14]Department of Botany, Faculty of Science, Palacky University, Olomouc, Czech Republic. [15]Sejet Planteforædling, Horsens, Denmark. [16]Nordic Seed, Odder, Denmark. [17]Natural Resources Institute Finland (Luke), Helsinki, Finland. [18]Institute of Biotechnology, University of Helsinki, Helsinki, Finland. [19]KeyGene, Wageningen, The Netherlands. [20]German Centre for Integrative Biodiversity Research (iDiv) Halle-Jena-Leipzig, Leipzig, Germany. [21]Center of Integrated Breeding Research (CiBreed), Georg-August-University, Göttingen, Germany. [22]Department of Agricultural Sciences, University of Helsinki, Helsinki, Finland. [23]Viikki Plant Science Centre, University of Helsinki, Helsinki, Finland, Córdoba, Spain. [24]Instituto Andaluz de Investigación y Formación Agraria, Pesquera, Alimentaria y de la Producción Ecológica (IFAPA), Área de Mejora y Biotecnología, Centro Alameda del Obispo, Córdoba, Spain. [25]Institute for Biological Data Science, CEPLAS, Heinrich Heine University Düsseldorf, Düsseldorf, Germany. [26]These authors contributed equally: Murukarthick Jayakodi, Agnieszka A. Golicz, Jonathan Kreplak. ✉e-mail: d.m.osullivan@reading.ac.uk; alan.schulman@helsinki.fi; sua@mbg.au.dk

## Methods

### Genome assembly and validation

PacBio HiFi reads were assembled using hifiasm v0.11-r302 (ref. [45]) with default parameters. The dovetail Omni-C data were aligned to the resulting contigs using minimap2 v2.20 (ref. [46]) to accurately order and orient the contigs. Similarly, the genetic markers from a consensus genetic map have been previously reported[47], and the 25K SNP array markers mapped in the NV644 × NV153 recombinant inbred lines (F6) were aligned to the preliminary contigs using minimap2 to assign contigs to chromosomes. Subsequently, the pseudomolecule construction was done with the TRITEX pipeline[48]. The final order and orientation of contigs in each chromosome were inspected and corrected manually with complementary support of Omni-C and the NV644 × NV153-derived genetic map. The assembled contigs were taxonomically classified using Kraken2 v2.1.1 (ref. [31]) with a database including sequences of plants, insects and bacteria, and with Blob-Tools v1.1 (ref. [49]). The genome completeness and consensus accuracy were evaluated by Merqury v1.3 (ref. [50]). The levels of homozygosity, heterozygosity and duplication were assessed by various tools such as Merqury, FindGSE v1.94 (ref. [21]) and GenomeScope v1.0 (ref. [51]). The centromere regions were identified in each chromosome using chromatin immunoprecipitation followed by sequencing (ChIP–seq) with the CENH3 (a centromere-specific histone H3 variant) antibody as previously reported[52]. In brief, the raw reads from ChIP–seq were trimmed by cutadapt v.1.15 (ref. [53]) and mapped to the preliminary pseudomolecules using minimap2. The alignments were converted to BAM format using SAMtools v1.15.1 (ref. [54]) and sorted by Novosort v3.06.05 (http://www.novocraft.com). The read depth was then calculated in 100-kb windows. Finally, the order of each chromosome was determined with regard to centromere positions (short-to-long arm), matching with the karyotype map of faba bean.

### Estimation of genome size using flow cytometry

Nuclear genome size was estimated by flow cytometry as previously described[55]. In brief, intact leaf tissues of the *V. faba* accession Hedin/2 and *Secale cereale* cv. Dankovske (2C = 16.19 pg DNA)[56], which served as the internal reference standard, were chopped together in a glass Petri dish containing 500 µl Otto I solution (0.1 M citric acid and 0.5% v/v Tween 20; Otto, 1990). The crude suspension was filtered through a 50-µm nylon mesh. Nuclei were then pelleted (300*g* for 2 min) and resuspended in 300 µl of Otto I solution. After 15 min of incubation on ice, 600 µl of Otto II solution supplemented with 50 µg ml$^{-1}$ RNase and 50 µg ml$^{-1}$ propidium iodide was added. Samples were analysed using a CyFlow Space flow cytometer (Sysmex Partec GmbH) equipped with a 532-nm green laser. The gain of the instrument was adjusted so that the peak representing G1 nuclei of the reference standard was positioned approximately on channel 100 on a histogram of relative propidium fluorescence intensity when using a 512-channel scale. The low-level threshold was set to channel 20 to eliminate particles with the lowest fluorescent intensity from the histogram; all remaining fluorescent events were recorded with no further gating used. Twelve Hedin/2 plants were sampled, and each sample was analysed three times, each time on a different day. A minimum of 5,000 nuclei per sample were analysed by the FloMax software (Sysmex Partec GmbH), and 2C DNA contents (in pg) were calculated from the means of the G1 peak positions by applying the formula: 2C nuclear DNA content = (sample G1 peak mean) × (standard 2C DNA content)/(standard G1 peak mean). The mean nuclear DNA content (2C) was then calculated for each species and DNA contents (in pg) were converted to the number of base pairs (in bp) using the conversion factor 1 pg DNA = 0.978 Gb (ref. [57]).

### Genome size estimation and quality assessment

The distribution of the *k*-mer (*K* = 101) frequency was estimated from PacBio HiFi reads using Jellyfish v2.2.10 (ref. [58]). The output histograms were used to estimate the genome size and heterozygosity using findGSE v1.94 (ref. [52]). The completeness of the assembly was assessed by two independent approaches: (1) self-alignment of HiFi reads to the assembly by minimap2 v2.20 followed by single variant (SV) calling using Sniffles v1.0.11 (ref. [59]); and (2) BUSCO v3.0.2b[60] analysis with Embryophyta database 9.

### Enzymatic methylation sequencing

DNA for methylome sequencing was extracted using the Qiagen DNEasy Plant 96 kit in accordance with the manufacturer's instructions, and checked for intactness on a 1% agarose gel and quantitated using the Thermo Fisher Quant-iT PicoGreen dsDNA Assay. Of Hedin/2 genomic DNA, 200 ng was combined with 0.001 ng of CpG-methylated pUC19 control DNA and 0.02 ng of unmethylated bacteriophage Lambda control DNA, then brought to a volume of 50 µl using EB buffer. The input DNA was sheared to 350–400 bp on the S220 Focused-Ultrasonicator Instrument (Covaris) using the following protocol: duty factor = 10; peak incident power = 175; cycles per burst = 200; time = 2 times 30 s. The sheared DNA was used to prepare a large insert NEBnext Enzymatic Methyl-seq library following the manufacturer's instructions (https://www.neb.com/-/media/nebus/files/manuals/manuale7120.pdf). Four libraries were constructed with different sequencing indexes. Index PCR was performed with five PCR cycles to include indexes and amplify the libraries. The final libraries were quantified by quantitative PCR, pooled at equimolar concentrations and sequenced for 500 cycles (2 × 250 bp paired-end reads) on an SP-flow cell of the Novaseq6000 system (Illumina).

### Tiffany genome assembly

The distribution of *k*-mers (*K* = 51) was estimated from PacBio HiFi reads using KAT v2.4.2 (ref. [61]). The output histograms were used to estimate genome size and heterozygosity using findGSE v1.94. Assembly was performed using hifiasm v0.15.5-r350. The completeness of the assembly was assessed by aligning HiFi reads back to contigs and calling structural variants using Sniffles v2.0.7. Despite there being no obvious heterozygous peak on the *k*-mer plots, we observed a higher proportion of BUSCO duplicate genes in Tiffany than in Hedin/2 and a slight overestimation of genome size with findGSE. In addition, we also noted a number of short contigs with read coverage about half of the expected, suggesting the presence of regions of heterozygosity in the otherwise mostly homozygous genome. We therefore performed haplotig purging using purge_haplotigs v1.1.2 (ref. [62]) (purge_haplotigs cov -l 3 -m 7 -h 25). The quality of the purged assembly was further evaluated using Merqury v1.3. Chromosome-level scaffolds were constructed with Rag-Tag v2.0.1 (ref. [63]) using the haplotig-purged assembly. To confirm the success of scaffolding, Hedin/2 and Tiffany chromosomes were aligned using GSAlign v1.0.22 (ref. [64]). We compared two approaches for Tiffany annotation to choose the one most suitable for comparative analyses: (1) individual annotation of genomes; and (2) a 'transfer and gap fill' approach (Supplementary Fig. 15). We observed that when genomes were individually annotated using the same pipeline, a proportion of syntenic genes had a different exon structure. These differences were substantially reduced when the Hedin/2 annotation was transferred onto Tiffany, suggesting that they might not reflect true biological differences. Artefactual differences in annotation, even when using the same pipeline, which could confound comparative analyses, have previously been reported[65,66]. We therefore used a transfer and gap fill approach, in which the Hedin/2 annotation was transferred onto Tiffany using Liftoff v1.6.1 (ref. [67]). To prevent the formation of chimeric gene models, caused, for example, by SVs, transferred models with in-frame stop codons were removed and replaced by Tiffany genes. Gene models unique to Tiffany were also added to the annotation. Overall, we observed that the transfer and gap fill approach resulted in more syntenic genes and more genes with the same coding sequence (CDS) length in both accessions.

## Repetitive DNA annotation

De novo repeat finding was done on Hedin/2 pseudomolecules with RepeatModeler v2.0.1 (ref. [68]) with sample sizes of 1,000,000 bp and with LTR_retriever v2.9.0 (ref. [69]) and LTRharvest[70]. De novo elements were clustered with cd-hit-est v4.8.1 (ref. [71]); element classification was aided by comparing to RepBase release 20181926, core-repeat domains from GyDB 2.0 (ref. [72]) and REXdb Viridiplantae v3.0 (ref. [73]). For the retrotransposons, sLTRs and full-length elements were specified as such by LTR_retriever and LTRharvest. Repeat masking was done with RepeatMasker v4.2.1 (http://www.repeatmasker.org) using de novo repeat libraries. Transposable element sequences encoding conserved protein domains were also identified based on their similarities to the REXdb v3.0 database using DANTE v0.1.1 (https://github.com/kavonrtep/dante). Satellite repeats were annotated using similarity searches to a custom database of satellite DNA families described in our previous studies[18,31,74,75]. The distribution of satellite repeats on metaphase chromosomes of *V. faba* was examined using fluorescence in situ hybridization (FISH). Chromosome preparation, probe labelling and FISH were performed as previously described[31], with hybridization and washing temperatures adjusted to account for the probe AT/CG content to allow for 10–20% mismatches. Chromosomes were counterstained with 4′,6-diamidino-2-phenylindole (DAPI), mounted in Vectashield medium (Vector Laboratories) and examined using a Zeiss AxioImager.Z2 microscope with an Axiocam 506 mono camera. Images were captured and processed using ZEN 3.2 software (Carl Zeiss).

## Gene model annotation

The repeat sequences were masked using RepeatMasker v4.1.1 (http://www.repeatmasker.org) with a custom repeat library generated by RepeatModeler v2.0.1 (using the Hedin/2 assembly). The gene annotation was conducted using BRAKER v2.1.6 (ref. [76]) (etpmode, min_contig 10000). The RNA sequencing libraries (Supplementary Table 2) were aligned using STAR 2.7.8a[77,78]. The protein database Viridiplantae OrthoDB v10.1 (ref. [79]) (https://onlinelibrary.wiley.com/doi/10.1111/tpj.14546merged), with the translated sequences of the previously published *V. faba* transcriptome assembly[16], was used as input for BRAKER, together with alignments generated by mapping the faba transcriptome assembly using GMAP v2020-10-14 (ref. [80]). In addition, *M. truncatula* genes ('Mt4.0v2_Genes') and *P. sativum* genes ('pissa.Cameor.gnm1.ann1.7SZR') were aligned using GMAP v2020-10-14. The generated alignments were used to polish the BRAKER gene models. To account for any gene models missed by BRAKER prediction but present in the Hedin/2 transcriptome assembly, the gene models from GMAP faba transcriptome alignments and BRAKER were compared using bedtools v2.30.0, retaining only the GMAP genes that did not have an intersection with the BRAKER gene models. For these genes, a further filtration was done to eliminate any short (less than 50 amino acids) translated proteins, in-frame stop codons or low (less than 200 reads) expression featureCounts, subread v2.0.1 (ref. [81]).

Completeness of the annotation was assessed for Hedin/2 and Tiffany by aligning one Iso-Seq dataset[82] and assembled transcriptomes produced for faba bean cultivars Hiverna, Dozah and Farah. Transcriptomes were mapped using GMAP v2020-10-14 and comparisons between those mappings and the annotations were made using bedtools[83]. Gene models that had been removed by polishing, but which intersected mapped transcripts, were rescued if the transcript was not a putative transposable element. *R* genes were detected on the unpolished and polished annotations using RGAugury v1.0 (ref. [84]). *R* genes present in the unpolished annotation but not in the polished annotation were also rescued. The coding potential for each transcript was computed with CPC2 v2.0 (ref. [85]). The mRNAs with low coding potential were reclassified as long non-coding RNAs. Genes of which at least 50% overlapped a transposable element domain were removed. Finally, any proteins that contained in-frame stop codons after phase

correction were also removed. The completeness of the final gene set was assessed using BUSCOv5.2.2 with the embryophyta_odb10 and fabales_odb10 databases.

## Symbiotic gene discovery

Total RNA sequencing was carried out for three biological replicates per condition. Eighteen libraries were prepared, and paired-end Illumina HiSeq mRNA sequencing (2 × 100 bp RNA sequencing) was performed by GeneWiz, which produced around 2 × 70 million reads per library on average. Adaptor sequences were removed using CLC Genomics Workbench 11 (CLC Bio Workbench, Qiagen). Only inserts of at least 30 nt were conserved for further analysis. Reads were mapped to the Hedin/2 genome using CLC Genomics Workbench 11 according to the manufacturer's recommendations. The mapped reads for each transcript were normalized as total counts and used for calculating gene expression. Intact and broken pairs were counted as one. The total counts of each transcript under different conditions were compared using proportion-based test statistics[86] implemented in the CLC Genomic Workbench suite. This β-binomial test compares the proportions of counts in a group of samples against those of another group of samples. Different weights were given to the samples, depending on their sizes (total counts). The weights were obtained by assuming a β-distribution on the proportions in a group, and estimating these, along with the proportion of a binomial distribution, by the method of moments. The result was a weighted *t*-type test statistic. We then calculated a false discovery rate correction for multiple-hypothesis tests[87]. Only genes with a minimum of ten reads for all compared conditions were considered in the analysis.

## Orthologous gene family identification

Genes from 19 legume species (Supplementary Table 6) were clustered to determine the orthologues relationship. The protein sequences from these species were aligned to each other using BLASTP v2.2.26 (ref. [88]) (-evalue $1 \times 10^{-5}$). The results were then used to cluster the gene families by OrthoMCL v2.0.9 (ref. [89]).

## Phylogenetic analysis and divergence time estimation

The single-copy genes identified from 19 legume species (Supplementary Table 6) by OrthoMCL v2.0.9 were selected for the phylogenetic analysis. The fourfold degenerate synonymous site (4d locus) was extracted to build the evolutionary tree by PhyML v3.0 (ref. [90]) and TreeBest v1.9.2 (https://github.com/Ensembl/treebest). Molecular clocks and divergence times were estimated using MCMCTREE v4.4 in the PAML v4.5 package[91] using the phylogenetic tree and the divergence time of known species (from published literature or using Timetree (http://www.timetree.org/)).

## Whole-genome duplication

The whole-genome duplication of *V. faba*, *M. truncatula* and *P. sativum* were estimated using the collinearity within each genome. First, synteny regions were identified using MCScanX v2.0 (ref. [92]). Then, the gene pairs in the synteny regions were used for 4dtv (fourfold degenerate transversions) calculation. The transversion rate was corrected by the HKY[93] model. The synonymous (Ks) and non-synonymous (Ka) substitutions were estimated by KaKs_Calculator v1.2 (ref. [94]).

## Tandem duplicate gene discovery

Tandem duplicated genes were also discovered using the CRBHits v0.0.4 package[95] function tandemdups. To confirm the results, genes were also classified using DupGen_finder v25Apr2019 (ref. [96]), with *A. thaliana* serving as the outgroup. *V. sativa* was excluded from TD analysis owing to suspected fragmentation of its structural annotation, which could result in inflation in the number of genes annotated as tandem duplications (TDs) (Supplementary Table 17). The age of duplications was estimated using $T = \mathrm{Ks}/2r$, $r = 1.5 \times 10^{-8}$. Ks was calculated using

CRBHits using method 'Li'. Synteny between Hedin/2 and Tiffany genes was analysed using CRBHits v0.0.4 package function rbh2dagchainer (type = 'idx', gap_length = 1, max_dist_allowed = 20), which internally uses the DAGchainer algorithm[97]. Syntenic tandem duplicated gene (TDG) clusters were discovered by connecting TDG clusters in individual genomes using the syntenic gene pairs found between Hedin/2 and Tiffany. To minimize the effect of unplaced genes on copy number variation analysis, as unplaced genes can result in spurious copy number variation calls, we corroborated the synteny-based results with Orthofinder v2.5.4 (ref. [98]) analysis. Only clusters that had the same or higher copy number in the same genotype, based on both synteny and Orthofinder results (for Orthofinder, only genes on the matching chromosomes and unplaced contigs were considered), were retained for further analysis. Syntenic clusters were functionally annotated with human readable descriptions using prot-scriber v0.1.0 (https://github.com/usadellab/prot-scriber).

## SPET library preparation and sequencing

Quantified genomic DNA using the Qubit 2.0 Fluorometer (Invitrogen) was used for library preparation, applying the Allegro Targeted Genotyping protocol (NuGEN Technologies), which relies on a panel of probes. Of DNA in solution, 20 ng μl$^{-1}$ was used as input following the manufacturer's instructions. All libraries were quantified using the Qubit 2.0 Fluorometer and library size was verified using the High Sensitivity DNA assay from Bioanalyzer (Agilent Technologies) or the High Sensitivity DNA assay from Caliper LabChip GX (Caliper Life Sciences). Libraries were also quantified by quantitative PCR, using the CFX96 Touch Real-Time PCR Detection System (Bio-Rad Laboratories). Samples were sequenced at IGA Technology Services (IGATech). DNA sequencing was performed on the Illumina NovaSeq 6000 (Illumina) in a 2 × 150 PE configuration, generating an average of 7.73 million sequenced read pairs per accession.

## Phenotyping and field trials

Seed traits were quantified using a MARViN seed analyzer (MARViTECH) on seeds harvested from trials at Sejet Plant Breeding, Sejet (55.82° N, 9.94° E) in 2019 (trial 23), 2020 (trial 26) and 2021 (trial 30), and at Nordic Seed, Dyngby (55.96° N, 10.25° E) in 2018 (trial 11), 2019 (trial 22) and 2020 (trial 25). Hilum colour was scored by visual inspection.

## SNP calling and GWAS

The SPET raw reads were trimmed with cutadapt v1.15 and aligned to the Hedin/2 genome using minimap2 v2.20. The alignments were sorted using Novosort v3.06.05 (http://www.novocraft.com), and BCFtools v.1.8 was used to call SNPs and short indels. The missing data in the VCF file were imputed using Beagle v5. The population structure analysis was performed with ADMIXTURE v1.3.0 (ref. [99]) with $K$ values ranging from two to ten. A fivefold cross-validation error for each $K$ was used to select the best $K$. The principle component analysis was performed using Plink v1.90b6.9 and the linkage disequilibrium (LD) blocks were identified using LDBlockshow v1.40. GWAS was performed with GEMMA v0.98.5 (ref. [100]), BLINK[101], FarmCPU[102] and EMMAX+EMMA200 (ref. [103]) using imputed SNP matrices. BLINK and FarmCPU were run using GAPIT3 v3.1 R package with three principal components. Only SNPs found by at least two methods were considered as candidates and it was further required that the signal was found in more than a single trial. The values used for the GWAS were the means of each genotype in each trial and the best linear unbiased estimator (BLUE). The BLUEs were computed with the lme4 package in R by first using the model:

$$y_{ijk} = \mu + G_i + E_j + G_i \times E_j + B_{jk} + \varepsilon_{ijk}$$

Where $y_{ijk}$ is the score of accession $i$ in environment $j$ in block $k$, $\mu$ is the overall mean of the trait, $G_i$ is the effect of accession $i$, $E_j$ is the effect of environment $j$, $G_i \times E_j$ is the interaction effect between accession $i$ and

environment $j$, $B_{jk}$ is the effect of block $k$ in environment $j$, and $\varepsilon_{ijk}$ is the residual. All effects except the mean were random effects. The significance of each random effect except $G$ were then tested one at a time using lmertest package in R. Only effects with a $P$ value larger than 0.05 were included in the final model. The final model had $G$ and $\mu$ as a fixed effect and all others as random. The BLUEs were then extracted from $G$.

Prediction accuracies of seed-size-related traits were investigated by fivefold cross-validation using the genomic best linear unbiased prediction (gBLUP) method. The fitted model in matrix notation is of the form $\mathbf{y} = \mathbf{1}\mu + \mathbf{Zu} + \mathbf{e}$, where $\mathbf{y}$ is a vector of observed phenotypic records (BLUEs), $\mu$ is the intercept, $\mathbf{1}$ is a vector of ones, $\mathbf{Z}$ is a design matrix linking records to accessions, $\mathbf{u}$ is a vector of (genomic) breeding values of the accessions assumed $\mathbf{u} \sim N(\mathbf{0}, \mathbf{G}\sigma_g^2)$, where $\sigma_g^2$ is the additive genetic variance and $\mathbf{G}$ is the genomic relationship matrix (GRM). The GRM was constructed as $\mathbf{G} = \mathbf{ZZ'}/2p_i(1 - p_i)$[104], where $\mathbf{Z}$ is the SNP matrix centred for the allele frequencies and $p_i$ is the allele frequency for the marker $i$. Finally, $\mathbf{e}$ is a vector of random residuals assumed $\mathbf{e} \sim N(\mathbf{0}, \mathbf{I}\sigma_e^2)$, where $\mathbf{I}$ is an identity matrix and $\sigma_e^2$ is the residual variance. Three prediction scenarios were investigated by varying the available markers for GMR calculation: (1) only candidate genome-wide association signals, (2) random samples of the same size as before repeated 100 times, and (3) all available SNP markers. Cross-validations were replicated ten times and averages ± standard deviations were reported. The 'mixed.solve' function from the rrBLUP v4.6.1 (ref. [105]) R package was used for all calculations.

## Premature termination codon and resistance gene analogue identification

SNPs and indels were filtered to retain biallellic variants only. Variants annotated as 'stop_gained' by SNPEff v4.3 (ref. [106]) were extracted, and only polymorphic variants with at least one homozygous reference and one homozygous alternative genotype were retained. Resistance gene analogues were identified using the RGAugury v1.0 pipeline[84]. Enrichment of premature termination codons in resistance gene analogues was calculated using the permTest function of regionerR v1.18.1 (ref. [107]) and 1,000 permutations (randomize.function=resampleRegions, evaluate.function=numOverlaps). All genes were provided as a universe for resampleRegions function.

## Identification of candidate gene for seed size

Positions of the most highly significant and stable SNPs associated with seed size were compared with positions of Hedin/2 protein-coding genes. Orthologues of the gene overlapping variant were identified using Orthofinder v2.5.4 (-M msa -S diamond -A mafft -T fasttree). Multiple sequence alignment of selected protein sequences was performed using Clustal Omega v1.2.4. The evolutionary history was inferred using the maximum likelihood method and JTT matrix-based model as implemented in MEGA X v10.2.6 (ref. [108]) with 100 bootstrap replicates. Publicly available expression data[16] for nine diverse tissues of Hedin/2 were used to quantify gene expression using Kallisto v0.44.0 (ref. [109]). LD patterns in the genomic interval surrounding the candidate gene were investigated using LDBlockShow v1.40 (ref. [110]).

## Hilum colour and histology

To examine hilum morphology, seed coat-containing hilum from inbred lines Hedin/2 (dark hilum) and Tiffany (pale hilum) were dissected from mature dry seed, saturated with 2% sucrose solution under vacuum for 1 h and embedded in cryo-gel media (Cryo-gel Leica). Samples were cut in cryotome (Leica CM1950) into 15-μm transversal sections and stained with Toluidine blue O (0.01%, w/v in water; Sigma Aldrich) as previously described[41,111]. Observation and photography were done on an Olympus BX 51 microscope (Olympus) in bright field, and figures were documented with an Apogee U4000 digital camera (Apogee Imaging Systems). For the investigation of metabolite content of surface layers of the hilum by laser desorption–ionization imaging mass spectrometry

(LDI-MS), seeds were mechanically cracked and hila with a small part of surrounding tissue were separated from the rest of seed coats using microscissors (MicroSupport), fixed using a double-sided tape on MALDI plates with outer surfaces facing up and analysed as previously described[41,112]. LDI-MSI experiments were done using a high-resolution tandem mass spectrometer (HRTMS) Synapt G2-S (Waters). The vacuum MALDI ion source used was equipped with a 350-nm 1-kHz Nd:YAG solid-state laser. Parameters of the mass spectrometer were set as follows: extraction voltage at 10 V, collision energies: trap collision energy (TrapCE) at 4 eV and transfer collision energy (TransferCE) at 2 eV. TrapCE at 25 eV and low mass (LM) resolution at 10 were used for MS/MS experiments. Instrument calibration was done using red phosphorus (1 mg.mL$^{-1}$, suspension in acetone). Mass imaging data collection was driven by HDImaging 1.5 software (Waters). The laser beam size was 60 μm. Spectra were collected in positive and negative ionization mode with laser energy at 300 arb. Laser repetition rate was set up at 1,000 Hz. Mass range was 50–1,200 Da. To fine-map the *HC* locus, a cross was made between inbred lines Disco (♀pale) and Hedin/2 (♂dark). F$_4$ seeds from 337 F$_3$ progeny of 21 F$_2$ individuals shown by flanking marker analysis to be heterozygous across the *HC* interval were scored for hilum colour, resulting in a 253 dark to 84 pale hilum ratio ($\chi^2 = 0.00098$, $P = 0.9749$ for fit to expected 3:1 ratio). A pool composed of equimolar quantities of DNA from each of the 84 recessive pseudo-F$_2$ individuals was created and subjected to SPET re-sequencing alongside DNA samples of the parent lines. To study expression of the PPO gene family in mid to late pod fill, individual plants of Hedin/2 and Tiffany were grown in the glasshouse until the most mature pods on lower nodes had almost reached maturity and the uppermost nodes were still in flower, giving a gradient of seed development. All pods were then harvested and dissected into pod wall, testa, cotyledon, funicle and embryo axis samples (Supplementary Fig. 16); fresh weights of each tissue were recorded. Because all pods on a given node are not fertilized synchronously and do not necessarily progress through development at the same rate, and on the basis of insights from our previous studies of faba bean seed development[113], we categorized individual pods into mid and late pod-fill stages in terms of the ratio of cotyledon weight to the total seed weight (Supplementary Fig. 17).

## PPO locus comparative sequence analysis

To identify PPO homologues in Hedin/2 and Tiffany proteomes, the protein sequence from the pea *PPO1/Pl* gene (*Psat1g206360*) was used as a BLAST v2.12.0 query. Multiple sequence alignment of PPO protein sequences was performed using Clustal Omega v1.2.4. The evolutionary history was inferred using the maximum likelihood method and JTT matrix-based model as implemented in MEGA X with 100 bootstrap replicates. The complete PPO regions (from the beginning of the first to the end of the last PPO gene and 10,000 bp flanking sequences on both sides) were extracted and aligned using minimap2 v2.24-r1122. Then, 20,000 bp downstream and upstream from the transcription start of *PPO-2* were extracted and similarity between sequences was visualized using FlexiDot v1.06 (ref. [114]).

## Gene expression analysis

RNA was extracted from 100 mg of flash-frozen dissected tissue (testa and cotyledon) using a Sigma Spectrum Kit (STRN250) according to the manufacturer's directions, except that incubation was made at room temperature after DNA digestion. Although extraction of RNA from cotyledons was performed exactly as per manufacturer's specifications, testa tissue was disrupted in an extraction buffer consisting of CTAB, PVP, 2 M Tris pH 8, 0.5 M EDTA pH 8, 4 M NaCl, spermidine and β-mercaptoethanol, followed by precipitation with 8 M lithium chloride (instead of the kit's lysis step). Total RNA was quantified using Qubit RNA IQ assay and normalized before preparation of directional mRNA sequencing libraries using standard methods. Between 4.1 and 5.6 million Illumina PE150 short reads per library (3× replicates, 2× tissues

and 2× genotypes) were generated. Hedin/2 and Tiffany gene expression were quantified using Kallisto v0.44.0 by pseudo-aligning RNA sequencing reads to respective reference transcripts. Transcript-level abundance was converted to gene-level abundance using tximport.

## Reporting summary

Further information on research design is available in the Nature Portfolio Reporting Summary linked to this article.

## Data availability

Raw data are available under European Nucleotide Archive study ID PRJEB52541. Genome assemblies and annotations for Hedin/2 and Tiffany are available for download at www.fabagenome.dk and can be accessed via an interactive genome browser (http://w3lamc.umbr.cas.cz/lamc/resources.html).

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

**Acknowledgements** This research was supported by grants from the Innovation Fund Denmark ('NORFAB', 5158-00004B) to J.S.; FACCE-JPI ERA-NET SusCrop Profaba to S.U.A., D.M.O., F.L.S. and A.H.S.; German Leibniz Association in the frame of the Leibniz Junior Research groups (J118/2021/REPLACE) to M.J.; the German Federal Ministry of Education and Research (de.NBI, 031A536B) to H.G.; Novo Nordisk Fund (#NNF20OC0065157) to A.H.S.; the Jane and Aatos Erkko Foundation grant (PanFaba) to A.H.S.; the Alexander von Humboldt Foundation in the framework of Sofja Kovalevskaja Award to A.A.G.; the Biotechnology and Biological Sciences Research Council award BB/P023509/1 to D.M.O.; the German Research Foundation (DFG) project number 497667402 to A.A.G. and B.U.; the Czech Science Foundation project number GACR 20-24252S to J.M.; the European Research Council under the European Union's Horizon 2020 research and innovation programme (grant agreement no. 834221) to J.S.; the BMBF-funded de.NBI Cloud within the German Network for Bioinformatics Infrastructure (de.NBI) (031A532B, 031A533A, 031A533B, 031A534A, 031A535A, 031A537A, 031A537B, 031A537C, 031A537D and 031A538A); and Grains Research Development Corporation as well as Agriculture Victoria for PacBio sequencing, parts of the public dataset curation for pulses. plantinformatics.io (DAV 1905-003RTX) and RNA sequencing data (BioProject ID PRJNA395480). We acknowledge CSC-IT Center for Science, Finland and ELIXIR CZ Research Infrastructure (Czech Ministry of Education, Youth and Sports grant no. LM2018131) for generous computational resources; and A. Fiebig for help with data submission.

**Author contributions** M.J. performed Hedin/2 genome assembly, methylation data analysis, hilum colour GWAS and wrote the first manuscript draft. A.A.G. performed Tiffany genome assembly, analysed seed coat and embryo expression data and compiled the premature termination codon atlas. A.A.G., J.K. and B.I. analysed synteny, tandem gene duplications and intron size variation. J.K., A.A.G. and L.I.F. contributed with gene annotation and data submission. S.U.A., D.M.O., L.I.F., D.A. and P.S. designed the SPET assay. D.A. and A.O.W. performed hilum colour bulk segregant analysis (BSA). P.B. carried out seed coat MS. E.B. and L.L.J. contributed with SPET data analysis and submission and prediction of seed size. P.-E.C. and R.B. generated and analysed rhizobial and mycorrhizal symbiosis RNA sequencing data. K.C. generated seed coat and embryo RNA sequencing libraries. J.D. and J.C. estimated genome size using flow cytometry. H.G., A.H.S., J.M., P.Neumann, P.Novák, J.T. and J.K. annotated and analysed repetitive elements. A.Hallab, P.T., B.U. and L.U.H. performed gene functional annotation. A.Himmelbach, S.P. and N.S. generated methylome sequencing data. S.K. and G.K.-G. contributed to PacBio sequencing, provided RNA sequencing data used for variant discovery, and integrated and visualized QTL and variation data online. A.K. and J.M. generated and analysed ChIP–seq data. L.K. and P.S. performed seed coat histochemistry. P.K. contributed seed coat analytical chemistry. T.W.M. carried out phenotype data validation and seed size GWAS. M.M., D.M.O., A.H.S., S.U.A., I.S., G.A., M.N. and H.K. edited the manuscript with input from all authors. J.O., L.K.N. and A.S. carried out SPET sample preparation and hilum colour phenotyping. P.Novák developed the faba bean genome browser. K.C. and T.R.-S.-H. prepared seed coat and embryo RNA sequencing libraries. L.A.R. contributed cytogenetic analysis. A.H.J.W. and R.J.S. generated HiFi data. H.Z. provided gene family, evolution, diversity and phylogenetic analysis. F.L.S. provided plant material. S.U.A., J.S., N.T. and A.M.T. provided resources. D.M.O. and D.A. genotyped pure Hedin/2 stock and generated the genetic map. S.U.A., A.H.S., D.M.O. and M.J. designed the study. S.U.A. coordinated the project.

**Competing interests** The authors declare no competing interests.

**Additional information**

**Correspondence and requests for materials** should be addressed to Donal Martin O'Sullivan, Alan H. Schulman or Stig Uggerhøj Andersen.

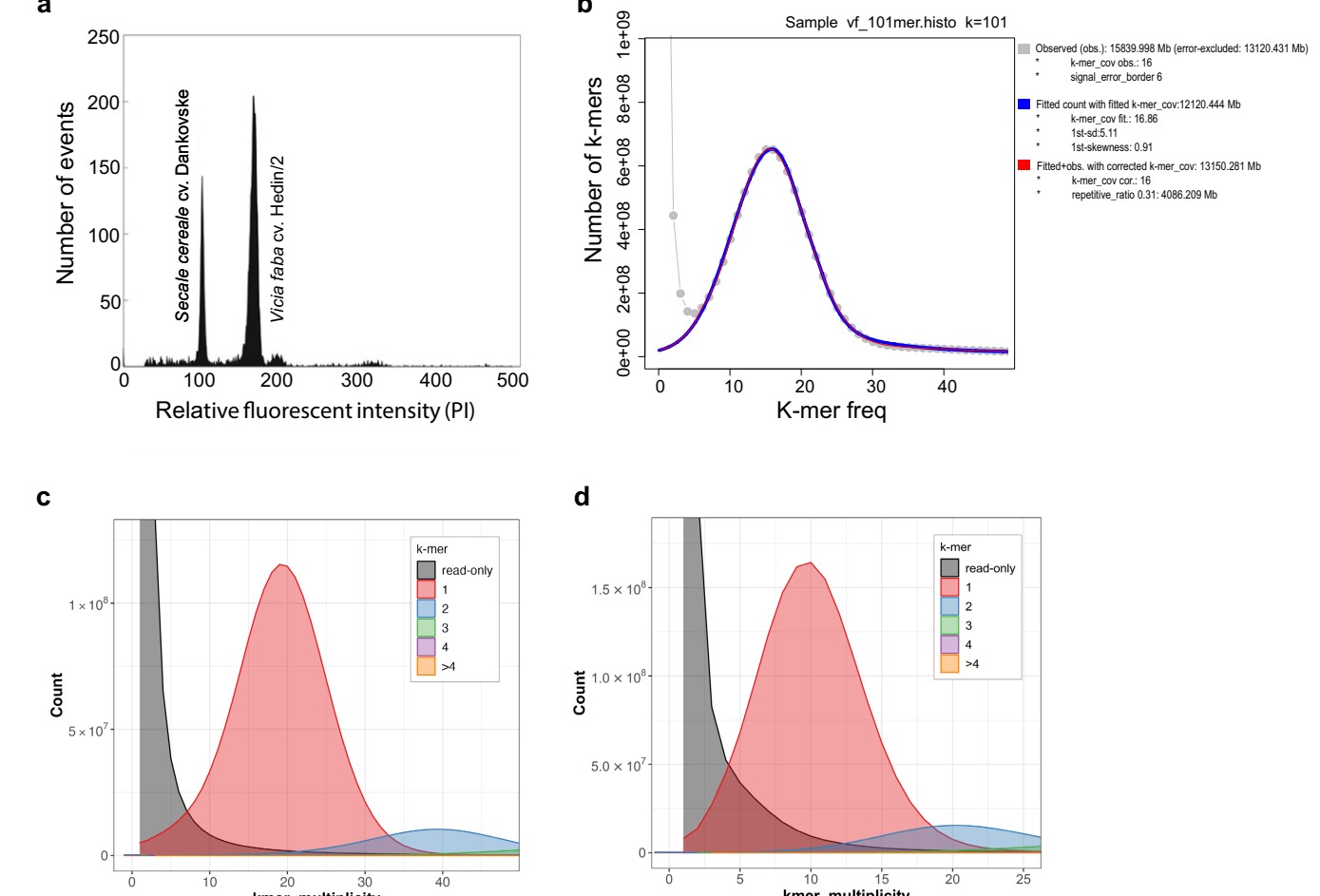

**Extended Data Fig. 1 | K-mer and flow-cytometry based genome size estimation. a**, Estimation of nuclear genome size of *V. faba* Hedin/2 using flow cytometry. The histogram of relative nuclear DNA content was obtained after flow cytometric analysis of propidium-iodide stained nuclei of *V. faba* 'Hedin/2' and *Secale cereale* cv. Dankovske (2C = 16.19 pg), which served as the internal reference standard. The low level threshold was set to channel 20 on a 512-channel scale histogram of relative fluorescence intensity; all remaining fluorescence events were recorded. 2C nuclear DNA content of *V. faba* Hedin/2 was estimated at 26.36 pg (± 0.26 s.d.). **b**, k-mer based estimation of genome size using Hedin/2 raw HiFi sequencing data. **c**, k-mer spectrum plot produced by Merqury for the Hedin/2 genome assembly. K-mer multiplicity refers to the number of times a k-mer is found in sequencing reads. Colour corresponds to the number of times a k-mer is found in the assembly. **d**, k-mer spectrum plot produced by Merqury for the Tiffany genome assembly.

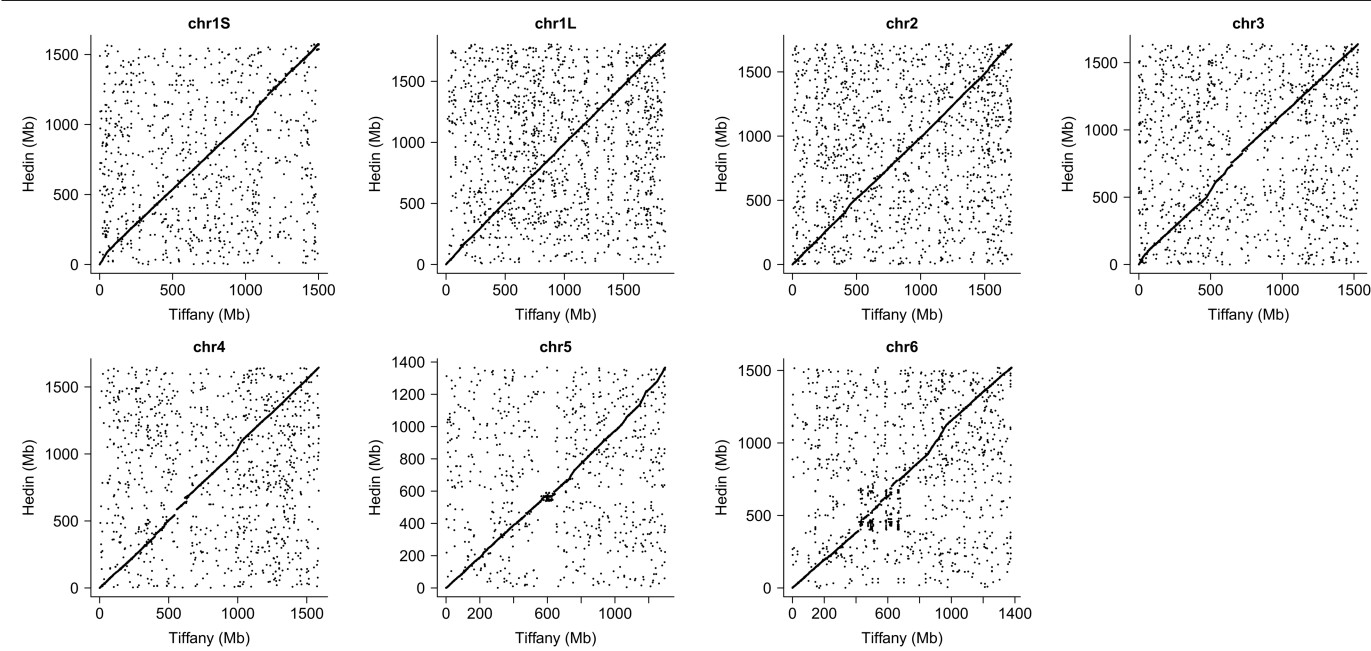

**Extended Data Fig. 2 | Hedin/2 and Tiffany chromosome alignments.** Dot plots showing alignments between Hedin/2 and Tiffany chromosomes.

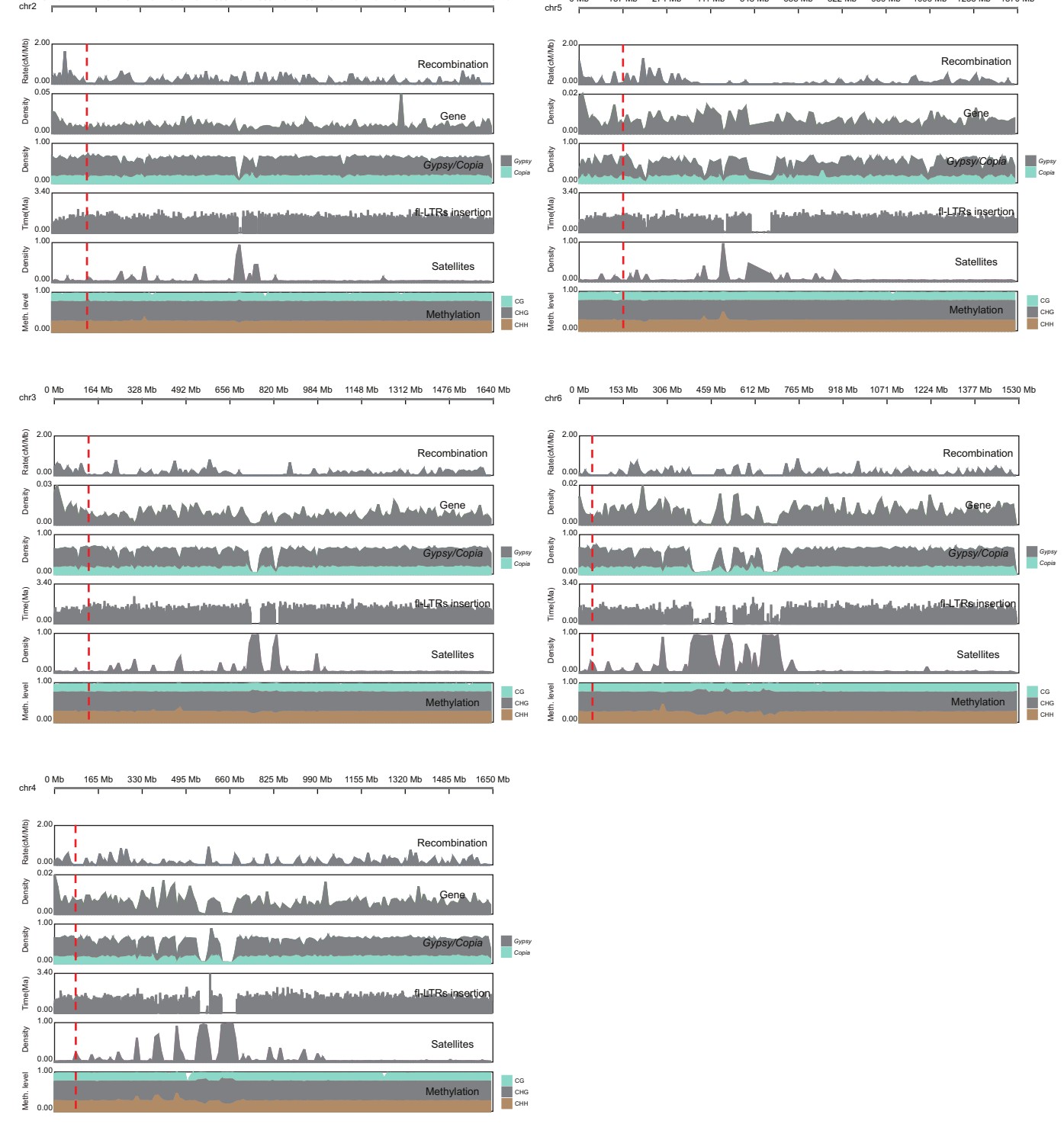

**Extended Data Fig. 3 | Genomic features of Hedin/2 chromosomes 2–6.**
Distribution of genomic components including recombination (cM/Mb),
gene density, retrotransposons of *Gypsy* and *Copia* superfamilies,
full-length (FL) LTR insertions, satellite repeats and DNA methylation
(CH, CHG and CHH context) on chromosomes 2 to 6.

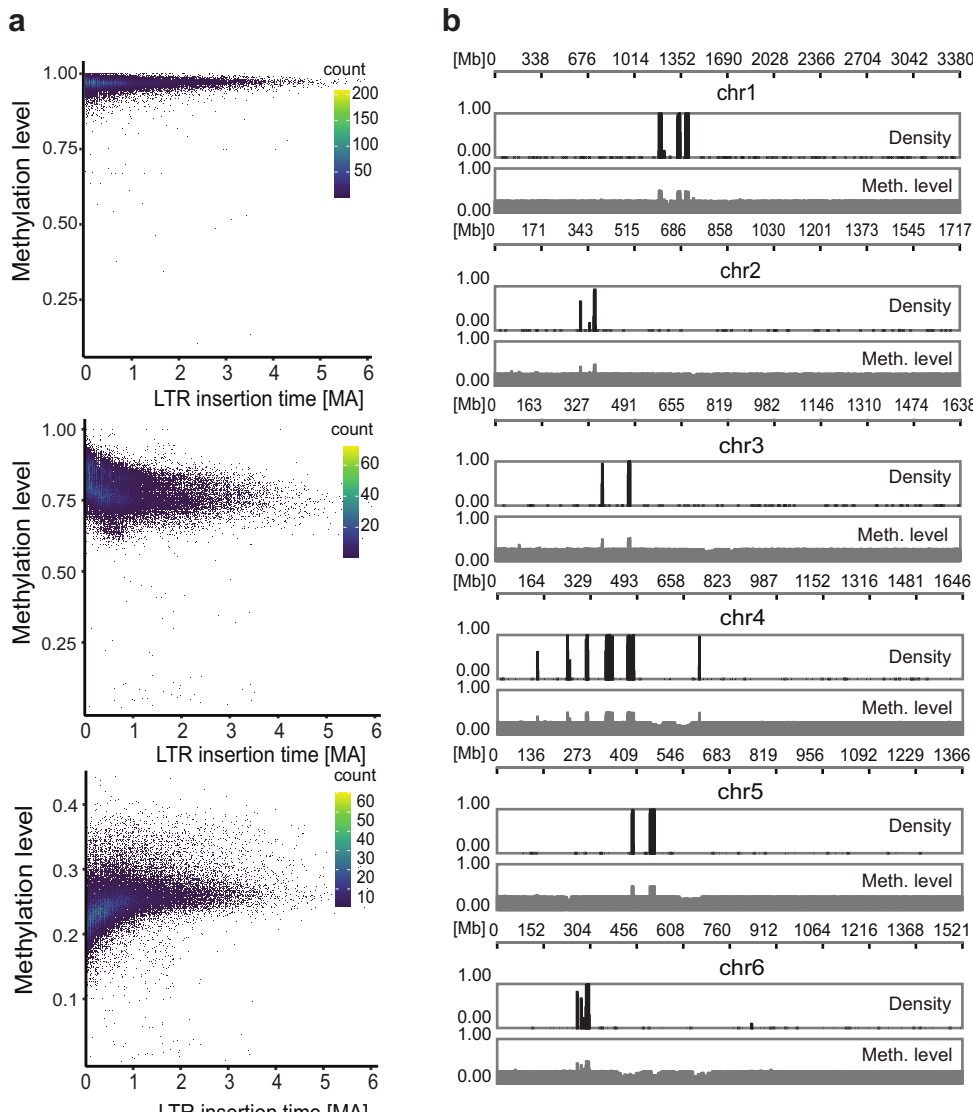

**Extended Data Fig. 4 | Methylation landscape in faba bean. a**, Landscape of CG (top), CHG (middle) and CHH (bottom) methylation with different TE insertion ages. **b**, CHH methylation peaks on FabTR-83 satellite repeats.

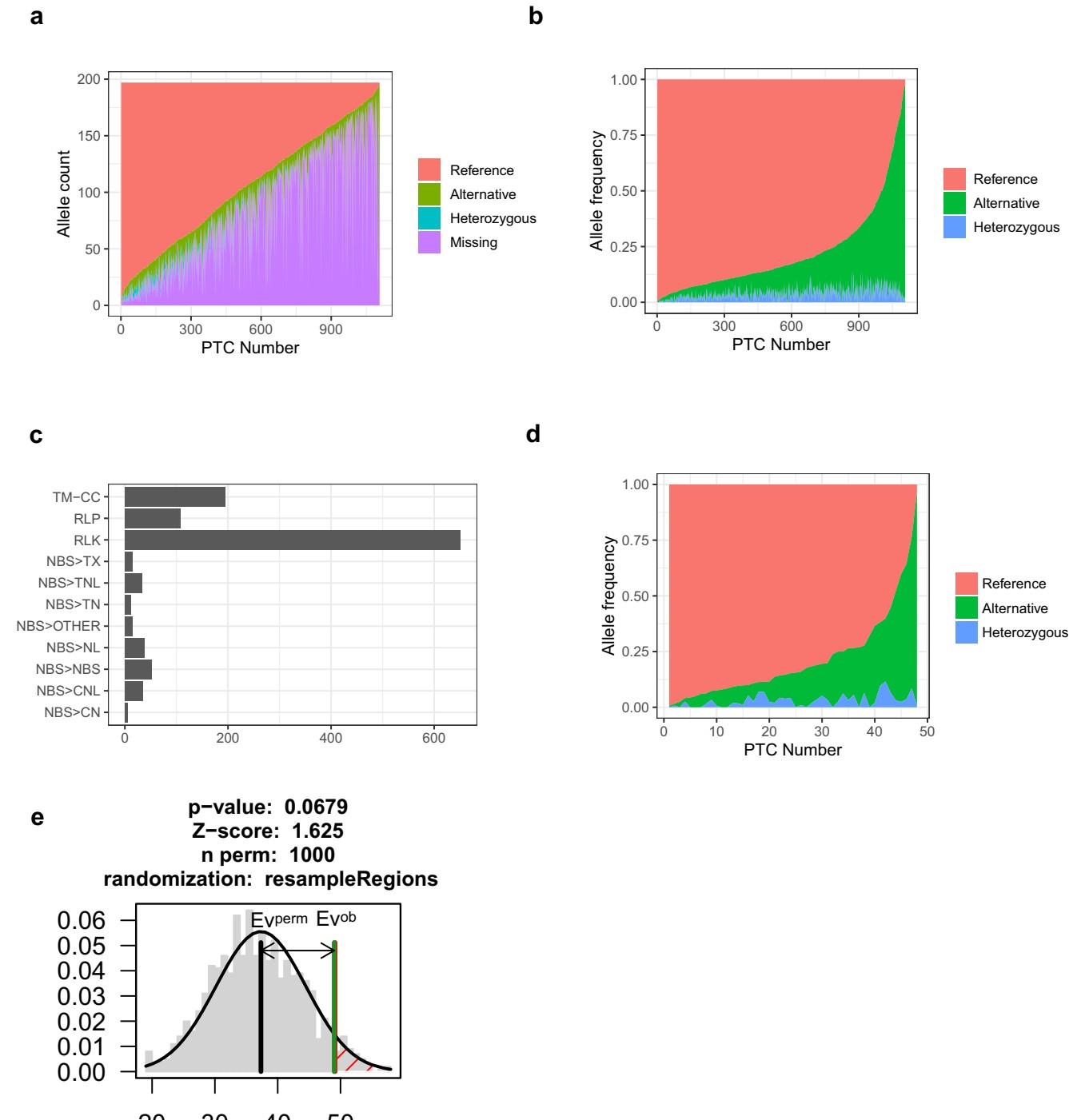

**Extended Data Fig. 5 | Variants resulting in premature termination codons.** Variants (SNPs and short InDels) resulting in premature termination codons (PTCs) identified from SPET data. **a**, Distribution of allele counts including missing genotype calls. **b**, Distribution of allele frequencies excluding missing genotype calls. **c**, Summary of resistance gene analogues (RGAs) identified in the Hedin/2 genome annotation. **d**, Distribution of allele frequencies of variants resulting in PTC in RGAs. **e**, Permutation test for over-representation of PTCs in RGAs.

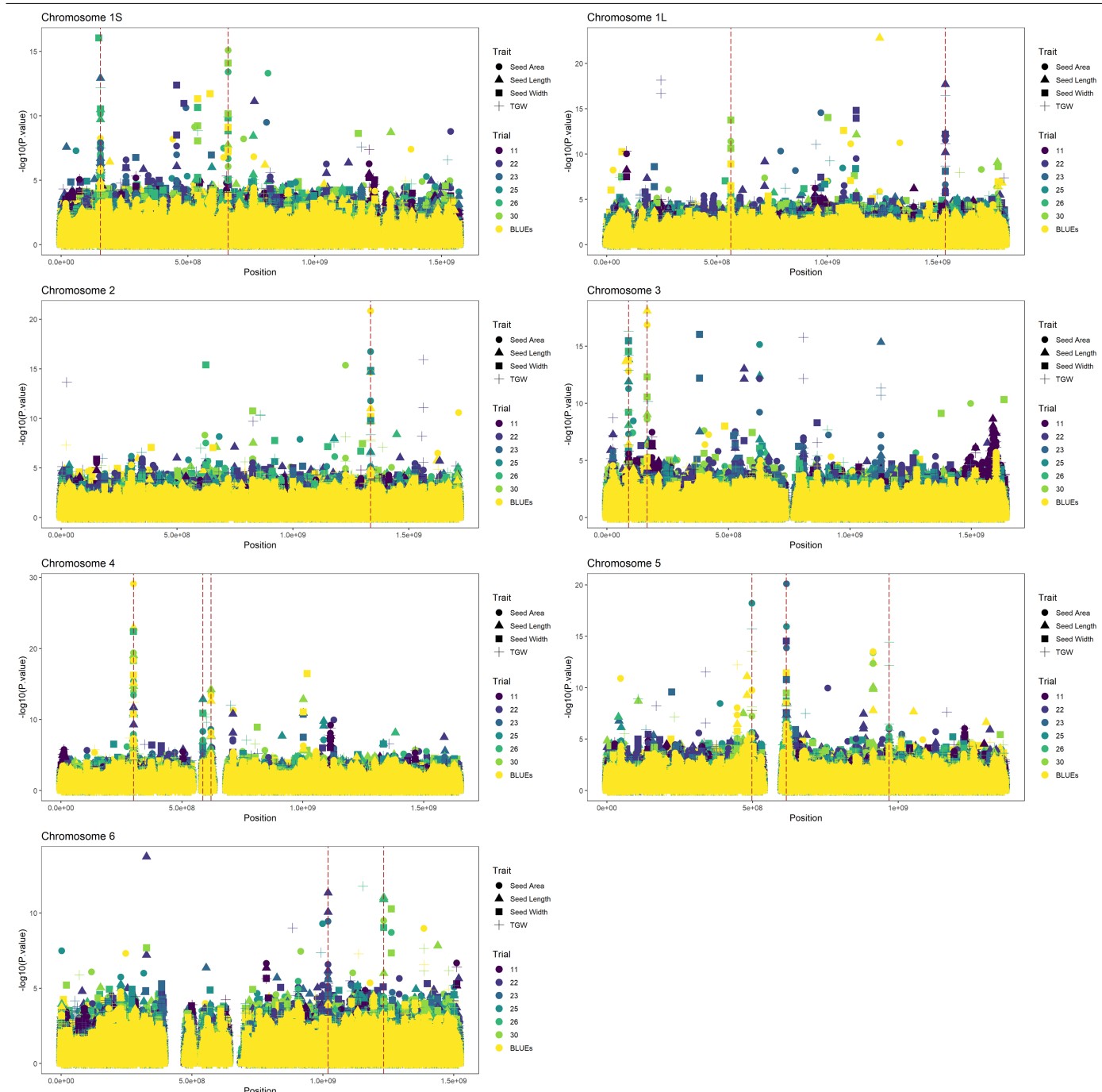

**Extended Data Fig. 6 | Manhattan plots of seed size GWAS.** Seed traits are shown as shapes. Trials are shown as colours. BLUE - best linear unbiased estimator. Candidate SNPs are shown as dashed lines.

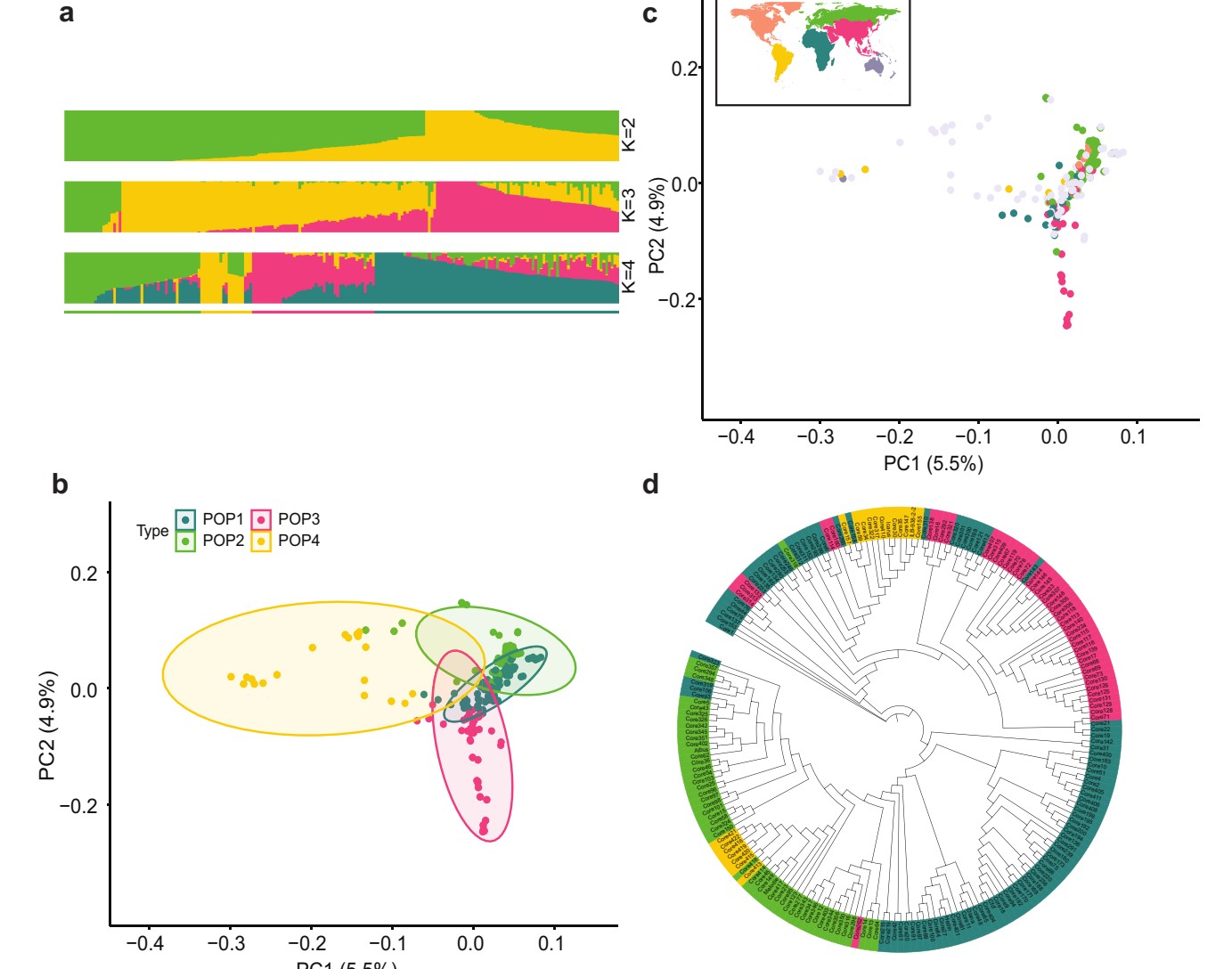

**Extended Data Fig. 7 | Genetic diversity and population structure.**
**a**, Population structure grouped by ADMIXTURE. **b**, Principal component analysis (PCA) of 197 accessions grouped with regard to subpopulations. **c**, PCA of 197 accessions coloured based on geographic origin. An African cluster extends to South America, a European cluster spans North America, an Asian cluster and other clusters are consistent with the observed ADMIXTURE result.

113 accessions (57.36%) could be assigned to a specific group, and 84 were identified as admixed resulting from hybridization between two or more of the four groups found with the analysis based on the membership coefficient (>=0.6). **d**, Neighbour joining (NJ) tree showing relationships among the 197 accessions.

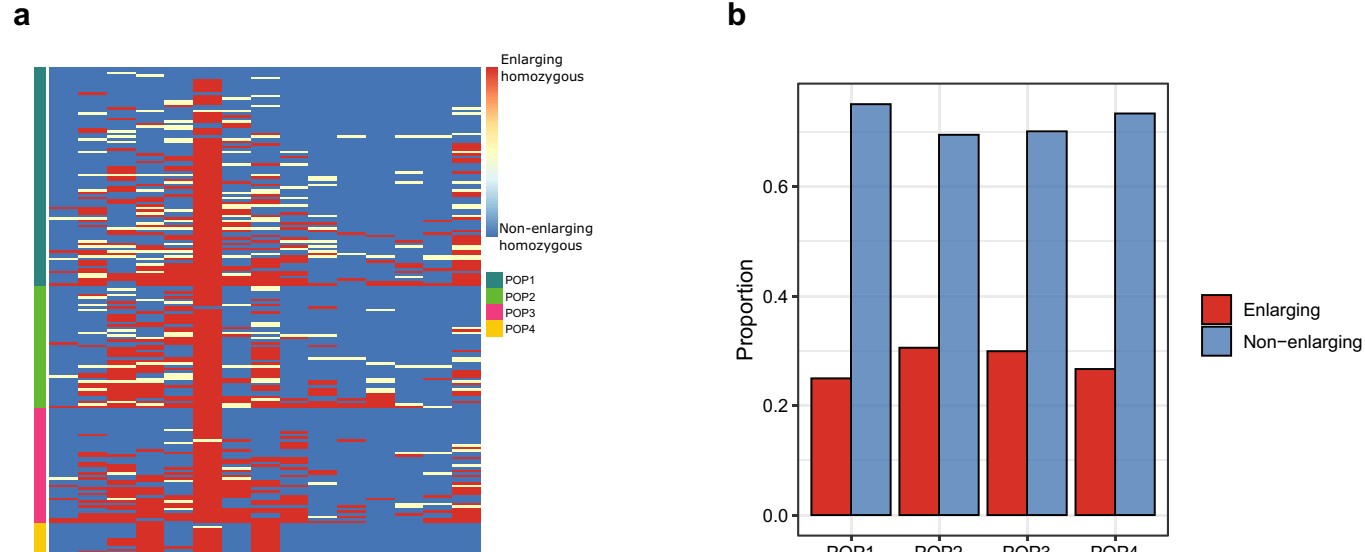

**Extended Data Fig. 8 | Seed enlarging alleles across populations. a**, distribution of seed enlarging alleles for 15 significantly associated SNPs (columns) across accessions and subpopulations (rows). **b**, Proportion of seed enlarging alleles in each subpopulation.

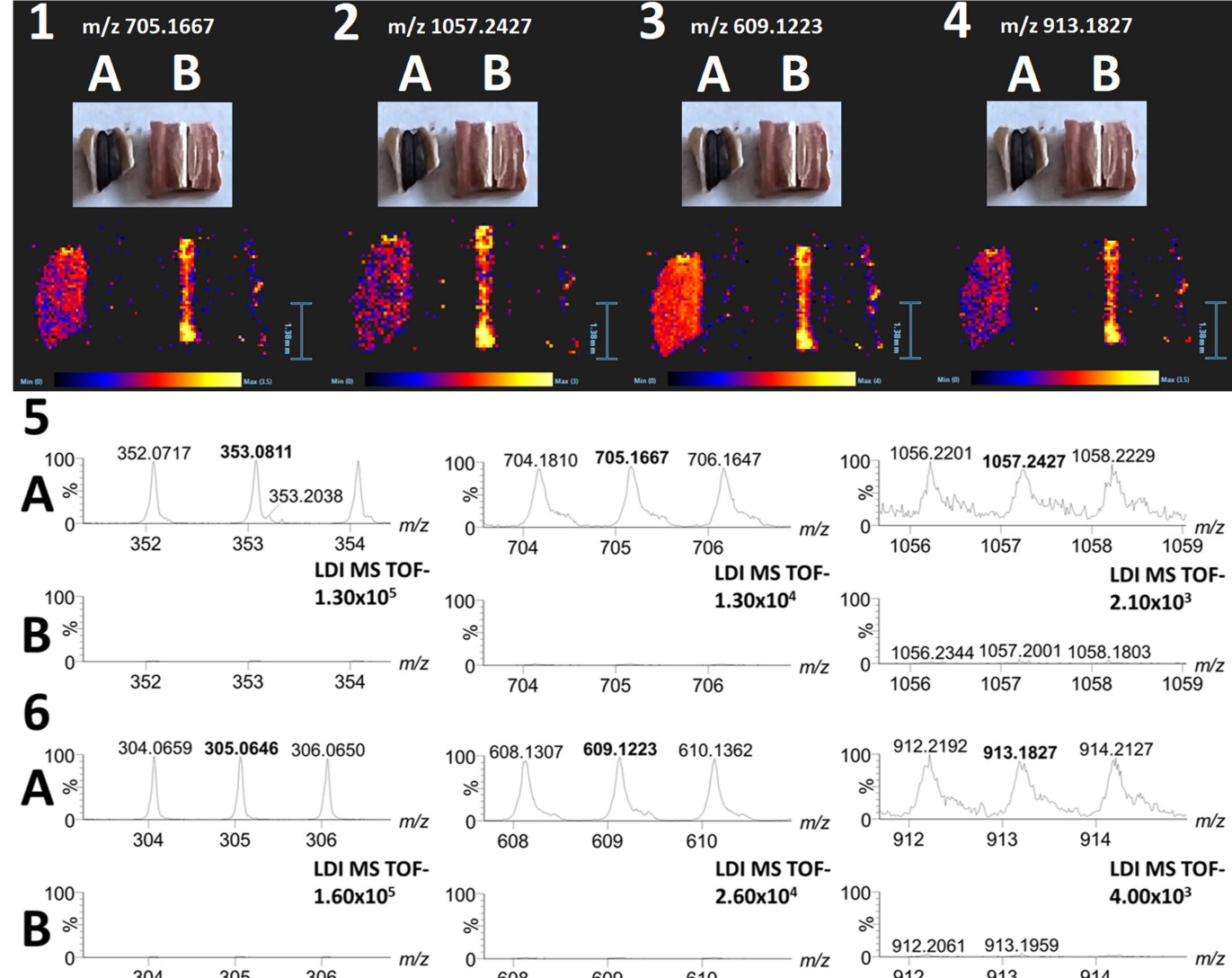

**Extended Data Fig. 9 | Phenolic compounds in faba bean hila.** Laser desorption-ionisation mass spectrometry imaging (LDI-MSI) showing polymerization of phenolic compounds associated with dark hilum colour in faba bean. Distribution of dimer and trimer of chlorogenic acid (1 and 2) and gallocatechin (3 and 4) on the surface of pigmented (A, Hedin/2) and nonpigmented (B, Tiffany) hila. Zoomed spectra of monomer, dimer and trimer of chlorogenic acid (5; signals at m/z 353.0811; 705.1667 and 1057.2427) and gallocatechin (6; signals at m/z 305.0646, 609.1223, 913.1827) collected from pigmented (A, Hedin/2) and non-pigmented (B, Tiffany) hila (the spectra showing particular signals are zoomed on the same intensity for both genotypes, A and B, e.g. the intensity 1.30x10⁵ is set for spectra 5A and 5B, etc.), spectra were collected from the compact surface without the hilar groove area and edges of the seed coat fragment. The identity of chlorogenic acid and gallocatechin dimers and trimers was further confirmed by characteristic fragments observed during related MS/MS experiment (i.e. anions of caffeic and chlorogenic acids cleaved from chlorogenic acid dimer and trimer at m/z 179.0341 and 353.0811, respectively, and products of retro-Diels Alder cleavage at m/z 179.0375 and 125.0232 from gallocatechin dimers and trimers, respectively).

**Extended Data Table 1 | Genome assembly and annotation statistics**

| | Hedin/2 | Tiffany<br>Purge Haplotigs<br>(no Purge Haplotigs) |
|---|---|---|
| Total assembly length | 11.9 Gb | 11.4 Gb |
| Total contig number | 10,721 | 14,378 |
| Contig N50 | 2.7 Mb | 1.6 Mb |
| Contig N90 | 737 Kb | 403 Kb |
| Longest contig length | 70 Mb | 25 Mb |
| Pseudomolecule size | 11.2 Gb | 10.9 Gb |
| Unanchored size | 648 Mb | 509 Mb |
| Genome completeness | 96.3% | 91.6% (95.8%) |
| Consensus quality value (QV) | 60.5 | 59.4 (57.6) |
| Number of gene models | 34,221 | 34,043 |
| Gene models with evidence of expression (>1TPM in at least one tissue) | 69% | N/A |
| Gene models with similarity to proteins of *V. faba* relatives | 93.3% | 93% |
| | | |
| Chromosome length (bp) | | |
| chr1 | 3,379,771,922 | 3,355,356,787 |
| chr2 | 1,716,769,615 | 1,709,916,750 |
| chr3 | 1,637,815,978 | 1,527,935,595 |
| chr4 | 1,645,877,737 | 1,588,398,909 |
| chr5 | 1,365,994,436 | 1,297,479,159 |
| chr6 | 1,520,236,431 | 1,379,031,673 |

Summary of genome assembly and annotation statistics for *V. faba* Hedin/2 and Tiffany genomes.

Alan H. Schulman
Stig U. Andersen

# Reporting Summary

## Statistics

For all statistical analyses, confirm that the following items are present in the figure legend, table legend, main text, or Methods section.

| n/a | Confirmed | |
|---|---|---|
| ☐ | ☒ | The exact sample size ($n$) for each experimental group/condition, given as a discrete number and unit of measurement |
| ☒ | ☐ | A statement on whether measurements were taken from distinct samples or whether the same sample was measured repeatedly |
| ☐ | ☒ | The statistical test(s) used AND whether they are one- or two-sided<br>*Only common tests should be described solely by name; describe more complex techniques in the Methods section.* |
| ☒ | ☐ | A description of all covariates tested |
| ☒ | ☐ | A description of any assumptions or corrections, such as tests of normality and adjustment for multiple comparisons |
| ☐ | ☒ | A full description of the statistical parameters including central tendency (e.g. means) or other basic estimates (e.g. regression coefficient) AND variation (e.g. standard deviation) or associated estimates of uncertainty (e.g. confidence intervals) |
| ☐ | ☒ | For null hypothesis testing, the test statistic (e.g. $F$, $t$, $r$) with confidence intervals, effect sizes, degrees of freedom and $P$ value noted<br>*Give P values as exact values whenever suitable.* |
| ☒ | ☐ | For Bayesian analysis, information on the choice of priors and Markov chain Monte Carlo settings |
| ☒ | ☐ | For hierarchical and complex designs, identification of the appropriate level for tests and full reporting of outcomes |
| ☐ | ☒ | Estimates of effect sizes (e.g. Cohen's $d$, Pearson's $r$), indicating how they were calculated |

*Our web collection on statistics for biologists contains articles on many of the points above.*

## Software and code

Policy information about availability of computer code

| Data collection | No software was used for data collection. |
|---|---|
| Data analysis | Multiple published software packages were used in the analysis including: ADMIXTURE v1.3.0, Beagle v5, bedtools v2.30.0, BLASTP v2.2.26, BLAST v2.12.0, BLINK (no version), BlobTools v1.1, BRAKER v2.1.6, BUSCO v3.0.2b, BUSCO v5.2.2, cd-hit-est v4.8.1, CLC Genomics Workbench 11, Clustal Omega v1.2.4, CPC2 v2.0, CRBHits v0.0.4, cutadapt v1.15, DANTE v0.1.1, DupGen_finder v25Apr2019, EMMAX (no version), FarmCPU (no version), featureCounts, subread 2.0.1, findGSE v1.94, FlexiDot v1.06, GAPIT3 v3.1, GEMMA v0.98.5, GenomeScope v1.0, GMAP v2020-10-14, GSAlign v1.0.22, hifiasm v0.11-r302, hifiasm v0.15.5-r350, Jellyfish v2.2.10, KaKs_Calculator v1.2, Kallisto v 0.44.0, KAT v2.4.2, Kraken2 v2.1.1, LDBlockShow v1.40, Liftoff v1.6.1, LTRharvest (no version), LTR_retriever v2.9.0, MCMCTREE v4.4, MCScanX v2.0, MEGA X v10.2.6, Merqury v1.3, minimap2 v2.20, minimap2 v 2.24-r1122, Novosort v3.06.05, Orthofinder v2.5.4, OrthoMCL v2.0.9, PAML v4.5, PhyML v3.0, prot-scriber v0.1.0, purge_haplotigs v1.1.2, regioneR v1.18.1, RepeatMasker v4.1.1, RepeatMasker v4.2.1, RepeatModeler v2.0.1, RGAugury v1.0, rrBLUP v4.6.1, SAMtools v1.15.1, Sniffles v1.0.11, Sniffles v2.0.7, SNPEff v4.3, STAR 2.7.8a, TreeBest v1.9.2, TRITEX pipeline (no version) |

For manuscripts utilizing custom algorithms or software that are central to the research but not yet described in published literature, software must be made available to editors and reviewers. We strongly encourage code deposition in a community repository (e.g. GitHub). See the Nature Portfolio guidelines for submitting code & software for further information.

## Data

Policy information about availability of data

All manuscripts must include a data availability statement. This statement should provide the following information, where applicable:

- Accession codes, unique identifiers, or web links for publicly available datasets
- A description of any restrictions on data availability
- For clinical datasets or third party data, please ensure that the statement adheres to our policy

Raw data are available under European Nucleotide Archive (ENA) study ID PRJEB52541. Genome assemblies and annotations for Hedin/2 and Tiffany are available for download at www.fabagenome.dk and can be accessed via an Interactive Genome Browser (http://w3lamc.umbr.cas.cz/lamc/resources.html).
Publicly accessible expression data used: PRJNA395480, SRS8798224-SRS8798245
Databases used in the study: BUSCO databases: embryophyta_odb9, embryophyta_odb10, fabales_odb10 (https://busco.ezlab.org/), REXdb v3.0 (http://repeatexplorer.org/?page_id=918), satDNA (10.1093/molbev/msaa090); Viridiplantae OrthoDB v10.1 (https://www.orthodb.org/), RepBase (https://www.girinst.org/repbase/, release 20181926), GyDB v2.0 (https://gydb.org/index.php?title=Main_Page)

## Human research participants

Policy information about studies involving human research participants and Sex and Gender in Research.

| | |
|---|---|
| Reporting on sex and gender | N/A |
| Population characteristics | N/A |
| Recruitment | N/A |
| Ethics oversight | N/A |

Note that full information on the approval of the study protocol must also be provided in the manuscript.

# Field-specific reporting

Please select the one below that is the best fit for your research. If you are not sure, read the appropriate sections before making your selection.

☒ Life sciences        ☐ Behavioural & social sciences        ☐ Ecological, evolutionary & environmental sciences

For a reference copy of the document with all sections, see nature.com/documents/nr-reporting-summary-flat.pdf

# Life sciences study design

All studies must disclose on these points even when the disclosure is negative.

| | |
|---|---|
| Sample size | The two accessions chosen for genome sequencing were selected based on their importance for the faba bean community. Hedin/2 was used for the development of genomic resources. Tiffany is a modern elite cultivar. They also have a contrasting hilum colour phenotype. Accessions for SPET genotyping were chosen to represent world-wide diversity. No sample size calculation was performed. |
| Data exclusions | No data was excluded. |
| Replication | Transcriptomics experiments (profiling of seed, root and nodule gene expression) were performed in triplicates. Field trials were performed across two locations: trials at Sejet Plant Breeding, Sejet (55.82°N, 9.94°E) in 2019 (trial 23), 2020 (trial 26), and 2021 (trial 30) and at Nordic Seed, Dyngby (55.96°N, 10.25°E) in 2018 (trial 11), 2019 (trial 22) and 2020 (trial 25). All attempts at replication were successful and replicates were used in the study. Where applicable, number of replicates is indicated in the methods and supplementary material. |
| Randomization | Randomization does not directly apply to genome sequencing and assembly studies. In cases where randomization procedures are part of computational analyses (for example bootstrapping used in phylogenetic inference) existing community standards were used. |
| Blinding | Study focuses on plant genome assembly and genomic analyses. The study design did not require and involve blinding. |

# Reporting for specific materials, systems and methods

We require information from authors about some types of materials, experimental systems and methods used in many studies. Here, indicate whether each material, system or method listed is relevant to your study. If you are not sure if a list item applies to your research, read the appropriate section before selecting a response.

## Materials & experimental systems

| n/a | Involved in the study |
|---|---|
| ☒ | ☐ Antibodies |
| ☒ | ☐ Eukaryotic cell lines |
| ☒ | ☐ Palaeontology and archaeology |
| ☒ | ☐ Animals and other organisms |
| ☒ | ☐ Clinical data |
| ☒ | ☐ Dual use research of concern |

## Methods

| n/a | Involved in the study |
|---|---|
| ☐ | ☒ ChIP-seq |
| ☐ | ☒ Flow cytometry |
| ☒ | ☐ MRI-based neuroimaging |

# ChIP-seq

## Data deposition

☒ Confirm that both raw and final processed data have been deposited in a public database such as GEO.

☐ Confirm that you have deposited or provided access to graph files (e.g. BED files) for the called peaks.

| | |
|---|---|
| Data access links *May remain private before publication.* | *For "Initial submission" or "Revised version" documents, provide reviewer access links. For your "Final submission" document, provide a link to the deposited data.* |
| Files in database submission | *Provide a list of all files available in the database submission.* |
| Genome browser session (e.g. UCSC) | *Provide a link to an anonymized genome browser session for "Initial submission" and "Revised version" documents only, to enable peer review. Write "no longer applicable" for "Final submission" documents.* |

## Methodology

| | |
|---|---|
| Replicates | *Describe the experimental replicates, specifying number, type and replicate agreement.* |
| Sequencing depth | *Describe the sequencing depth for each experiment, providing the total number of reads, uniquely mapped reads, length of reads and whether they were paired- or single-end.* |
| Antibodies | *Describe the antibodies used for the ChIP-seq experiments; as applicable, provide supplier name, catalog number, clone name, and lot number.* |
| Peak calling parameters | *Specify the command line program and parameters used for read mapping and peak calling, including the ChIP, control and index files used.* |
| Data quality | *Describe the methods used to ensure data quality in full detail, including how many peaks are at FDR 5% and above 5-fold enrichment.* |
| Software | The centromere regions were identified in each chromosome using ChIP-seq with the CENH3 (a centromere-specific histone H3 variant) antibody reported by Avila Robledillo L et al. Briefly, the raw reads from the ChiP-seq were trimmed by cutadapt (v.1.15) and mapped to the preliminary pseudomolecules using minimap2. The alignments were converted to BAM format using SAMtools and sorted by Novosort (V3.06.05) (http://www.novocraft.com). The read depth was then calculated in 100 kb windows. |

# Flow Cytometry

## Plots

Confirm that:

☒ The axis labels state the marker and fluorochrome used (e.g. CD4-FITC).

☒ The axis scales are clearly visible. Include numbers along axes only for bottom left plot of group (a 'group' is an analysis of identical markers).

☐ All plots are contour plots with outliers or pseudocolor plots.

☐ A numerical value for number of cells or percentage (with statistics) is provided.

## Methodology

| | |
|---|---|
| Sample preparation | Nuclear genome size was estimated by flow cytometry as described previously (Doležel et al., 2007). Briefly, intact leaf tissues of a V. faba accession Hedin/2 and Secale cereale cv. Dankovske (2C = 16.19 pg DNA; Doležel et al. 1992), which served as the internal reference standard, were chopped together in a glass Petri dish containing 500 μl Otto I solution (0.1M citric acid, 0.5% v/v Tween 20; Otto, 1990). The crude suspension was filtered through a 50 μm nylon mesh. Nuclei were then pelleted (300 × g, 2 min) and resuspended in 300 μl of Otto I solution. After 15 min of incubation on ice, 600 μl of Otto II solution supplemented with 50 μg/ml RNase and 50 μg/ml propidium iodide were added. |
| Instrument | Samples were analyzed using a CyFlow Space flow cytometer (Sysmex Partec GmbH, Görlitz, Germany) equipped with a 532 |

| Instrument | nm green laser. The gain of the instrument was adjusted so that the peak representing G1 nuclei of the standard was positioned approximately on channel 100 on a histogram of relative fluorescence intensity when using a 512-channel scale |
|---|---|
| Software | Analysis was performed using FloMax software (Sysmex Partec GmbH, Görlitz, Germany) and 2C DNA contents (in pg) were calculated from the means of the G1 peak positions by applying the formula: 2C nuclear DNA content = (sample G1 peak mean) × (standard 2C DNA content) / (standard G1 peak mean). The mean nuclear DNA content (2C) was then calculated for each species. DNA contents in pg were converted to genome size in bp using the conversion factor 1 pg DNA = 0.978 Gbp (Doležel et al., 2003). |
| Cell population abundance | 12 individual Hedin/2 plants were sampled, and each sample was analyzed three times, each time on a different day. A minimum of 5000 nuclei per sample were analyzed. |
| Gating strategy | The low level threshold was set to channel 20 to eliminate particles with the lowest fluorescent intensity from the histogram, all remaining fluorescent events were recorded with no further gating used |

☐ Tick this box to confirm that a figure exemplifying the gating strategy is provided in the Supplementary Information.

