## [Peer Review File · Nature]

Manuscript Title: The giant diploid faba genome unlocks variation in a global protein crop

Reviewer Comments & Author Rebuttals

Reviewer Reports on the Initial Version:

Referee expertise:

Referee #1: plant evolutionary genetics

Referee #2: legume genetics

Referee #3: plant genomics

Referees' comments:

Referee #1 (Remarks to the Author):

A. Summary of the key results

=====

The manuscript titled "The giant diploid faba genome unlocks variation in a global protein crop" describes several important results for this species. They are summarized as:

- The authors present the reference genome for the species *Vicia faba* accession "Hedin/2" with a genome size of 13 Gb. The assembly size was 11.9 Gb with 94% of the assembly anchored in 6 chromosomes. The centromeric regions were identified by chromatin immunoprecipitation sequencing for centromeric histone H3, as well as the FISH for three repetitive element families. The authors also sequenced and assembled a different variety named "Tiffany" for which they obtain a similar quality.
- 34,221 and 34,043 gene models were predicted for the "Hedin/2" and the "Tiffany" genomes respectively with a BUSCO completeness percentage of 96%. Although the big genome size of this species, it has a high ratio of collinearity with other legumes. No extra WGD besides the papilionoid 55 MYA duplication were found, so the extremely expanded genome is associated with a transposable element burst.
- The *V. faba* genome presented an important expansion on the LTR Ogre transposable element accounting for 44.4% of the genome. TE density agreed with the recombination rate across the chromosomes. The more recent TE burst was dated ~1 MYA. The analysis of the high methylated state of the genome revealed that TE burst was not associated with a low methylation status.
- A SPET panel with 90,000 probes was designed with re-sequencing data and used to perform a

GWAS with a diversity panel of 197 accessions.

- The author found two candidate genes in tandem (VfPPO-2 and VfPPO-3) associated with hilum color. VfPPO-2 gene expression supported its involvement in this trait, although two other genes (VfPPO-6 and VfPPO-7) were upregulated in the pale hilum genotype (Tiffany).

B. Originality and significance: if not novel, please include reference

=====
The manuscript is original and due the importance of the species as crop and its big genome, this could be considered a genomic milestone for this species. Nevertheless, putting this work in the context of other plant genomic manuscript that I reviewed lately, I found some studies missing. They are at the stage of pan-genomes and they are identifying several agronomical traits with large GWAS panels. In this regards this study may have a limited significance. I also found missing more information about the population study of this manuscript even if the authors described the GWAS for 197 accessions, but I also understand that there are limitations of space for this type of manuscript.

In overall terms, I think that the development of this genome is an important resource for the community and in that sense it is significant and worthy of publishing, specially for a complex genome as this one, but I think that the authors could include more "biological" information (for example derived from the GWAS analysis or mining the genome for other molecular sources for the TE burst beyond the methylation analysis).

C. Data & methodology: validity of approach, quality of data, quality of presentation.

=====
=====

Due the brevity of the manuscript it is difficult to evaluate some parts. There are some missing QC evaluations that should be performed to have a better idea of the quality of the genome assembly. Please check the guidelines supplied by the Earth Biogenome Project (<https://www.earthbiogenome.org/assembly-standards>), but here some examples:

- Genome completeness. It was evaluated only in term of the gene space (BUSCO). Merqury (or KAT or any other Kmer method) should be used also to estimate the completeness. Read re-mapping could be also a good idea. In this sense, it is also good to evaluate the number of complete LTR elements with the LAI (LTR_retriever).
- Consensus accuracy. Although HiFi data has a low error rate, HiFi assemblies are not error safe. The heterozygosity may drive to the production of chimeric kmers that do not exist in the reads. Tools as merqury can help to identify those.
- Duplicated regions. Tools like purge_dups or purge_haplotigs and Merqury can help to assess this.
- Contaminations. Assessed with Blobtools, specially for small contigs.

D. Appropriate use of statistics and treatment of uncertainties

=====

Overall, the use of the statistics of the works presented in this manuscript is appropriate to my knowledge.

E. Conclusions: robustness, validity, reliability

=====

The conclusions are robust.

F. Suggested improvements: experiments, data for possible revision

=====

a. Biological insights about the TE ogre burst. I think that one of the most interesting questions that post this genome is, why it had this TE ogre burst ~1MYA, specially compared with closed related species like *V. sativa*. I think that the authors should explore the genes space for genes associated with TE control (e.g., see for example the studies about the mPing TE in rice) comparing them with other species where they do not have this genomic trait.

b. Insufficient assembly and annotation QC: See the section C, but mostly I recommend running Merqury to assess assembly completeness and consensus accuracy, purge_dups to assess possible duplicated regions. Some supplementary data for the Hi-C experiment is missing (coverage, contacts...). Some information about the estimated heterozygosity will be useful too. About the annotation, it will be good to know how many gene models were supported by experimental data (RNA-Seq and close related protein sequences).

c. Quality assessment different between both accessions. SV were called with Sniffles for "Hedin/2" and CuteSV for "Tiffany" which could drive to a bias to compare both assemblies.

d. The "transfer and gap filling approach" is not clear/convincing to me. I think that may drive to chimeras. It is expected to find SV between both genomes, so if they are not described, the use of the information of a genome to "complete" the other one can drive to produce chimeras. Probably the use of pan-genome tools like VG and the visualization of the graphs can help with this rather than assume a fully syntenic order for all the genes.

e. Missing data in several sections:

i. There are some data (e.g., FISH data at the Figure 1) from which no description could be found in the main text (Results and Material and Methods).

ii. It will be useful for the community to have the whole sequence of the probe for the Affx ID of the Suppl. Table 1. Otherwise, this data can't be re-checked and reused.

iii. The results about population structure are limited to two sentences and one extended figure. No information about the accessions in a table, or what it is represented here. Are all the accessions modern cultivars or there are also some wild/ancestral/heirloom types?

iv. Poor information supplied for the GWAS. Although the authors used the hilum color to illustrate the results of the GWAS, I think that results should be presented in the Suppl. Tables (accessions, phenotypes for these accessions...).

f. I think that it will be a good idea to contextualize the transposable element expansion for closed related genomes, such as *Vicia sativa* and *Pisum sativum*. This could be done with a violin plot with x-axis associated with TE class and species, and y-axis for TE age (e.g., <https://bmcplantbiol.biomedcentral.com/articles/10.1186/s12870-021-02858-1/figures/3>).

g. Comparison with the *V. sativa* genome is missing. Although the *V. sativa* genome quality may not be as high as this one, I think that the manuscript should have a small paragraph describing the comparison between both genomes.

h. Discussion section is more of a conclusion than a discussion (it may be a switch in the header of the section).

G. References: appropriate credit to previous work?

=====

I think so. I understand that the number of citations may be limited in this manuscript. Nevertheless, it will be useful to read in the introduction that the *Vicia sativa* genome was already published and cite according.

H. Clarity and context: lucidity of abstract/summary, appropriateness of abstract, introduction and conclusions

=====

=====

The main text has several part missing that will help with the manuscript reading. Right now, I have the feeling of a lot of work and results compressed into a short manuscript what drive to many questions during its reading. The abstract does not contain information about genome assembly size, number of gene models, and candidate genes associated with the presented trait. The introduction has missing that the *Vicia sativa* genome was published. Probably this type of information could be useful for the reader. I think that the header of the Conclusion section has been mistaken by Discussion (that it is included in the results).

Reviewed by Aureliano Bombarely on October 31st, 2022.

Referee #2 (Remarks to the Author):

Jayakodi and colleagues report a complete genome for the fava bean, an important grain legume. With the support of deep coverage with long reads and Hi-C, the authors can capture nearly 12GB of the approximately 13GB diploid genome. The authors show how transposable elements have expanded rapidly in the lineage leading to fava beans. A complete genome also facilitates an understanding of the evolution of gene families in comparison to other legumes, and patterns of

tandem duplication. More important for future breeding, the authors develop dense genomewide markers for trait discovery.

This is a really important paper. Fava bean has immense potential for increasing food and nutritional security and increasing the availability of plant protein as a highly productive cool season legume. Furthermore, this work is a technical achievement, due to the size and complexity of the fava bean genome.

The abundance of high-quality genomic data is presented clearly.

An example of the clear presentation of results is the mapping of hilum color, an important agronomic characteristic. This trait is linked (by 5-10cM) to vicine levels but matters on its own as a quality trait (for reasons that are not well explained, despite the otherwise clear presentation). Statistical analyses are appropriate for a genome paper, including assembly, TE evolution, diversity analysis, and GWAS.

I have a handful of suggestions for minor improvement of the manuscript.

Some unique aspects of fava bean are not mentioned until the discussion, such as vicine and favism, or the lack of a compatible wild relative and the challenges this poses for understanding fava bean domestication. This may confuse some readers.

The authors have worked on hilum color many years. It is not clearly explained by this trait is agronomically significant. It is a great example trait, as the complexity of the repeat cluster harboring the GWA hit would be impossible to resolve without a high quality genome. PPO analysis of metabolites in seeds with varying hilum color needs further explanation.

Fava is unusual due to a lack of compatible wild relative. It is not clear if a wild relative simply has not been found, if the immediate ancestor is extinct, or if sufficient genome evolution occurred in the lineage leading to cultivated fava bean that it became incompatible with other *Vicia* species such as the phenotypically similar *Vicia narbonensis*. This unique aspect of fava is not mentioned until the discussion.

As is typical of genome papers at the highest tier journals, the authors have placed far more data and results than can be presented in the space allowed. This is to be expected, but some of the more interesting results are buried in supplemental tables and barely described in the methods. Expanding a few legends will help without great expansion of text. For example, pretzel visualizations and examination of further traits are barely mentioned in the main text. The figure legend for it, and the methods, could allow some description.

In the same vein, geographic regional differentiation could be better explained. This is something that the supplemental figure and legend might address at more length. The four regions do vary in their patterns of genetic variation. This may give some insight into where fava originated, or how it has shifted with historical migration along with human groups.

Referee #3 (Remarks to the Author):

This is an extremely well written manuscript describing the assembly, annotation, and characterization of the faba bean genome. The novelty of the entire manuscript is that this is a huge genome. The authors did a thorough analysis of the genome and overall, it is interesting in how it differs from other large plant genomes that are polyploid or diploid with high repetitive sequence content. I perhaps think this is the most fascinating aspect of the genome that receives limited attention in the manuscript.

Overall the authors performed standard genome analyses to document the quality of their genome (which is high) and how it can be used to identify genes involved in key traits via the example of hilum color in two varieties. The methods are well described and the results clearly shown and interpreted.

The Introduction is highly abbreviated thus a naïve reader will not understand much about faba bean. I do think its positive attributes are overly stated in the Introduction as all of the US and some of lower Canada is soybean production thus I do not believe temperate regions of Europe and America can not yield sufficient plant-based protein. The problem is that the majority of soy is not for human consumption, instead it is animal feed and byproducts.

Colors in Fig 1d difficult to see. Suggest change the maroon and blue with brighter colors to contrast with the dark grey. Explain what the red dotted line is, I assume it is the centromere.

The different levels/phases of annotation in Extended data table 3 are confusing. Add a legend to explain these.

Line 236, unclear why Fig 1d is cited.

More details on the data availability of the genome and annotation is warranted.

A statement of why Hedin/2 was selected as the initial reference genome is warranted.

A detailed analysis of gene location vs recombination location would have been very engaging.

A statement of how (or if) faba bean can be transformed should be added.

A deeper description of the diversity panel would help learn what genetic bottlenecks there are for faba bean.

Author Rebuttals to Initial Comments:

Referee #1 (Remarks to the Author):

A. Summary of the key results

=====

The manuscript titled “The giant diploid faba genome unlocks variation in a global protein crop” describes several important results for this species. They are summarised as:

- The authors present the reference genome for the species *Vicia faba* accession “Hedin/2” with a genome size of 13 Gb. The assembly size was 11.9 Gb with 94% of the assembly anchored in 6 chromosomes. The centromeric regions were identified by chromatin immunoprecipitation sequencing for centromeric histone H3, as well as the FISH for three repetitive element families. The authors also sequenced and assembled a different variety named “Tiffany” for which they obtain a similar quality.
- 34,221 and 34,043 gene models were predicted for the “Hedin/2” and the “Tiffany” genomes respectively with a BUSCO completeness percentage of 96%. Although the big genome size of this species, it has a high ratio of collinearity with other legumes. No extra WGD besides the papilionoid 55 MYA duplication were found, so the extremely expanded genome is associated with a transposable element burst.
- The *V. faba* genome presented an important expansion on the LTR Ogre transposable element accounting for 44.4% of the genome. TE density agreed with the recombination rate across the chromosomes. The more recent TE burst was dated ~1 MYA. The analysis of the high methylated state of the genome revealed that TE burst was not associated with a low methylation status.
- A SPET panel with 90,000 probes was designed with re-sequencing data and used to perform a GWAS with a diversity panel of 197 accessions.
- The author found two candidate genes in tandem (VfPPO-2 and VfPPO-3) associated with hilum color. VfPPO-2 gene expression supported its involvement in this trait, although two other genes (VfPPO-6 and VfPPO-7) were upregulated in the pale hilum genotype (Tiffany).

B. Originality and significance: if not novel, please include reference

=====

The manuscript is original and due the importance of the species as crop and its big genome, this could be considered a genomic milestone for this species. Nevertheless, putting this work in the context of other plant genomic manuscript that I reviewed lately, I found some studies missing. They are at the stage of pan-genomes

and they are identifying several agronomical traits with large GWAS panels. In this regards this study may have a limited significance. I also found missing more information about the population study of this manuscript even if the authors described the GWAS for 197 accessions, but I also understand that there are limitations of space for this type of manuscript.

In overall terms, I think that the development of this genome is an important resource for the community and in that sense it is significant and worthy of publishing, specially for a complex genome as this one, but I think that the authors could include more "biological" information (for example derived from the GWAS analysis or mining the genome for other molecular sources for the TE burst beyond the methylation analysis).

Answer: Thank you for these suggestions. We have carefully considered how to add most value and impact to the manuscript and have prioritised GWAS analysis of seed size and its implications for faba bean population genetics. We have also mined the genomes for possible molecular sources for the TE burst as suggested, but found this less informative. Please see the revised manuscript and below for the full details.

C. Data & methodology: validity of approach, quality of data, quality of presentation.

=====

Due the brevity of the manuscript it is difficult to evaluate some parts. There are some missing QC evaluations that should be performed to have a better idea of the quality of the genome assembly. Please check the guidelines supplied by the Earth Biogenome Project (<https://www.earthbiogenome.org/assembly-standards>), but here some examples:

- *Genome completeness. It was evaluated only in term of the gene space (BUSCO). Merqury (or KAT or any other Kmer method) should be used also to estimate the completeness. Read re-mapping could be also a good idea. In this sense, it is also good to evaluate the number of complete LTR elements with the LAI (LTR_retriever).*
- *Consensus accuracy. Although HiFi data has a low error rate, HiFi assemblies are not error safe. The heterozygosity may drive to the production of chimeric kmers that do not exist in the reads. Tools as merqury can help to identify those.*
- *Duplicated regions. Tools like purge_dups or purge_haplotigs and Merqury can help to assess this.*
- *Contaminations. Assessed with Blobtools, specially for small contigs.*

Answer: We very much agree that quality control is essential for ensuring delivery of excellent genomic resources. We have now implemented additional quality control measures as suggested. Please see section F. b. below for the full details.

D. Appropriate use of statistics and treatment of uncertainties

=====

Overall, the use of the statistics of the works presented in this manuscript is appropriate to my knowledge.

E. Conclusions: robustness, validity, reliability

=====

The conclusions are robust.

F. Suggested improvements: experiments, data for possible revision

=====

a. Biological insights about the TE ogre burst. I think that one of the most interesting questions that post this genome is, why it had this TE ogre burst ~1MYA, specially compared with closed related species like V. sativa. I think that the authors should explore the genes space for genes associated with TE control (e.g., see for example the studies about the mPing TE in rice) comparing them with other species where they do not have this genomic trait.

Thank you for the question; we agree that control of retrotransposon replication regarding genome size in *V. faba* compared to related species is of high interest.

The suggestion of the referee to look at the gene space of *V. faba* and *V. sativa* for genes involved in suppression of retrotransposons is a good one, regarding whether is any obvious difference in mechanisms that might suppress propagation of *Ogre* and other retrotransposons in the extant genomes. We have compiled a set of genes with identified roles in transcriptional silencing by RNA-directed DNA methylation (RdDM) and in post-transcriptional silencing by miRNA-directed cleavage processes. Our survey of the two genomes, as well as of four other sequenced legume genomes including the close relatives *Pisum sativum* and *Lens culinaris*, found that no gene encoding a known component of a silencing pathway was missing from *V. faba*. Consistent with these results, we had seen that global levels of methylation, as reported in the manuscript for the CG, CHG and, notably for retrotransposons, the CHH context was very high; transcriptional silencing in the current *V. faba* appears to be fully functional. We have included the cross-species comparison of TE-control genes as an Extended data table and refer to it in the text.

There are two aspects that impact *Ogre* accumulation - control of replication and mechanism of loss.

With respect to control of replication, retrotransposons are well known to be transcriptionally activated by stresses such as drought. What might have been the activating forces prevailing 0.5 MYA is difficult to say since the direct wild ancestor of *V. faba* is unknown, although it was likely somewhere in the Fertile Crescent. The region currently has a N-S aridity gradient, though evidence indicates it was earlier wetter. The peak of prevalence at 0.5 MYA holds both for *Gypsy* and *Copia* elements, families of which have different promoters and likely non-identical activating pathways. Likewise, the range of the ancestor of *V. sativa* is unknown. A hunt for clues in the *V. sativa* genome in comparison to that of *V. faba* would be interesting, but the former has a contig N50 of only one-tenth of the latter. Strikingly, in barley (Mascher et al. 2021 doi:10.1093/plcell/koab077), a large population of young retrotransposons was only seen in the genome when a high-quality long-read assembly was made, as it is precisely the young elements that are incorporated poorly into highly gapped assemblies. In order to properly evaluate the retrotransposon populations of the two *Vicia* species, their genomes should be sequenced to the same contiguity and then assembled and annotated with the same pipelines and parameters. In conclusion, the different qualities of the *V. sativa* and *V. faba* assemblies currently prevent us from gaining further insight into the origins of the *Ogre* burst by genomic comparisons of these species.

Regarding loss, examination of earlier work (Macas et al. 2015, doi:10.1371/journal.pone.0143424), where the same approach was taken across several *Vicia* spp., sheds some light on the question. The work found *Ogre* covering 22.5% of the *V. sativa* genome and 54.3% of *V. faba*. In the same work, the ratio of solo-LTR to full-length elements was estimated at 1.6 for *V. sativa* and 0.7 for *V. faba*. This indicates that loss by LTR : LTR recombination has been much more effective for *Ogre* in *V. sativa* than in *V. faba*. Interestingly, this difference was also seen for the *Gypsy Chromovirus* family, but not for *Athila* or for the *Copia* retrotransposons, indicating that it is not solely a general property of recombination in these species. We have now referred to the Macas paper to highlight differences between *V. sativa* and *V. faba*.

Unfortunately, we cannot draw much inspiration from the *mPing* system for the *Vicia* story, as *mPing* is a non-autonomous DNA transposon, a MITE, mobilised by transposase from the autonomous *Pong* in rice, and moving by a cut-and-paste mechanism that is unlike the copy-and-paste of retrotransposons. In *V. faba*, DNA transposons represent less than 0.6% of the genome. Suppression mechanisms for DNA transposition, which doesn't involve an RNA phase and reverse transcription, is likewise only partially overlapping with that for retrotransposons.

b. Insufficient assembly and annotation QC: See the section C, but mostly I recommend running Merqury to assess assembly completeness and consensus accuracy, purge_dups to assess possible duplicated regions. Some supplementary data for the Hi-C experiment is missing (coverage, contacts...). Some information about the estimated heterozygosity will be useful too. About the annotation, it will be good to know how many gene models were supported by experimental data (RNA-Seq and close related protein sequences).

Answer: We appreciate the reviewer's concern and have addressed the genome assembly validation at multiple level as follows:

Genome completeness : First, we manually corrected the chimeric contigs and structural errors, such as inversion, using Hi-C (Omni-C) data and a genetic map (Monat et al. 2019; <https://doi.org/10.1186/s13059-019-1899-5>). Because BUSCO has limitations in evaluating the most difficult-to-assemble regions of the genome, as the reviewer suggested, we used Merqury (Rhie et. al. 2020; <https://doi.org/10.1186/s13059-020-02134-9>), which uses k-mers to assess the genome in a reference-free manner, to validate our assembly. The Merqury results indicate that the completeness of the genome reached 96.3% for the Hedin/2 de novo assembly, verifying the genome completeness (**Extended Data Table 1**). Further, the LAI score (10.5) indicates the continuity of our reference genome assembly. Similarly, we provided supporting data to validate our reference-guided 'Tiffany' assembly (**Extended Data Table 1**).

Consensus accuracy: Yes, we are aware that Hifiasm produces chimeric contigs, though very few. We manually identified and corrected them using Hi-C. Furthermore, we estimated the assembly consensus quality by Merqury; we obtained a consensus quality value (QV) of 60.5 for the Hedin/2 assembly (**Extended Data Table 1**), indicating a more accurate consensus in our assembly. For the haplotig purged Tiffany assembly, the QV was 59.4

Duplicated regions: We performed contig assembly using Hifiasm, which can purge duplications between haplotigs without relying on third-party tools such as purge_dups. Further, one would be able to observe an unexpected assembly size that might result from diverged haplotypes or heterozygosity. For Hedin/2, we did not observe such an unusual pattern in our contig assembly. Further, the k-mer spectrum plots generated with Merqury showed no abnormal false duplications in our genome assembly. In addition, for an independent check, a separate analysis with GenomeScope showed a heterozygosity of 0.33% and duplication of 0.37% in the Hedin/2 assembly. These results strongly support a high homozygosity and absence of duplications. For the Tiffany assembly, we did observe an inflated assembly size and used the Purge Haplotigs pipeline to remove ~10% of the sequence, mostly on short contigs. After purging, the gene duplication level (as evaluated by BUSCO) was very similar for Tiffany and Hedin/2, further supporting the absence of duplicated haplotigs in the Hedin/2 assembly.

Contaminations: We have screened for DNA contaminants in all HiFi contigs and unanchored small contigs (< 1 Mb) using Kraken2 and Blobtools (reviewer suggested tool) respectively. No evidence of contamination with foreign DNA from a different taxon was detected in the assembly. In addition, we removed plastid contigs in our assembly after aligning to the mitochondrial and chloroplast genomes. For Tiffany, contigs with abnormal coverage (potential contaminants) were removed during the Purge Haplotigs run.

Annotation: For the Hedin/2 annotation, 69% of gene models were supported by RNA-Seq data (expression >1TPM in at least one tissue as evaluated by Kallisto using a panel of nine diverse tissue types). In addition, 93.3% and 93% of gene models for Hedin/2 and Tiffany had similarity to proteins of close relatives (pea, lentil, medicago, diamond, e-value cutoff 1e-5). We updated Extended Table 1 accordingly.

c. Quality assessment different between both accessions. SV were called with Sniffles for "Hedin/2" and CuteSV for "Tiffany" which could drive to a bias to compare both assemblies.

Answer: We re-called SVs with Sniffles and obtained very similar results. Less than 100 SV >1 kb in length for the whole genome were obtained, confirming the high quality of the genome already reflected in Merqury assessments.

d. The “transfer and gap filling approach” is not clear/convincing to me. I think that may drive to chimeras. It is expected to find SV between both genomes, so if they are not described, the use of the information of a genome to “complete” the other one can drive to produce chimeras. Probably the use of pan-genome tools like VG and the visualization of the graphs can help with this rather than assume a fully syntenic order for all the genes.

Answer: We considered issues such as SV and chimeras during the process of developing the most appropriate annotation strategy for comparative analyses. When we performed individual annotation of the Hedin/2 and Tiffany genomes we noticed quite a lot of genes in syntenic positions, but which had slightly different exon structures. The differences disappeared when annotation transfer from Hedin/2 to Tiffany was done. We consulted with Mario Stanke, whose lab develops the BRAKER pipeline, and he confirmed that also they observed that effect and that the differences in gene structure are likely artifacts. Comparative annotation approaches are being developed to combat the issue (<https://www.ncbi.nlm.nih.gov/pmc/articles/PMC6028123/>), however those rely on full genome Progressive Cactus alignments which for now are not tested in faba bean due to its unique repeat profile. We are actively working on solving this issue, keeping in mind future pangenome studies. For now, to avoid artefactual differences, we use the transfer and gap-fill approach. Annotation transfer has previously been used in pangenome studies, for example barley (doi.org/10.1038/s41586-020-2947-8). We transfer only genes whose ORF remains intact after transfer. Genes, which after transfer have an in frame STOP (which often happens due to SV presence), are removed and replaced with Tiffany models. Transferred annotation was also supplemented with Tiffany-specific genes. When we use the transfer and gap fill approach, we observe more syntenic genes and more genes with the same gene structure between Hedin/2 and Tiffany (the number of genes with a different CDS length is reduced by half). While this approach may miss some Tiffany-specific features, it avoids the issue of artefactual differences and makes the annotations more comparable. Additionally, for all the key results presented (for example PPO gene cluster analysis) we performed the analysis in parallel for both genotypes and transferred annotations to ensure the conclusions are robust and not affected by the chosen annotation strategy. Unfortunately, graphical pangenomics tools are also not yet tested on faba bean. We are also actively working on this (<https://github.com/pangenome/pggb/issues/187>).

e. Missing data in several sections:

i. There are some data (e.g., FISH data at the Figure 1) from which no description could be found in the main text (Results and Material and Methods).

Answer: Thanks for pointing this out. We have added the details to the Methods section.

ii. It will be useful for the community to have the whole sequence of the probe for the Affx ID of the Suppl. Table 1. Otherwise, this data can't be re-checked and reused.

Answer: Thanks for the valuable suggestion. We have now included the probe sequences in **Supplementary table 1**.

iii. The results about population structure are limited to two sentences and one extended figure. No information about the accessions in a table, or what it is represented here. Are all the accessions modern cultivars or there are also some wild/ancestral/heirloom types?

Answer: All of our accessions are cultivated and spring type. We have now added the passport information for each accession in **Supplementary table 11**.

iv. Poor information supplied for the GWAS. Although the authors used the hilum color to illustrate the results of the GWAS, I think that results should be presented in the Suppl. Tables (accessions, phenotypes for these accessions...).

Answer: We have now updated **Supplementary table 11** with primary passport information for our diversity panel and phenotype data for five traits (hilum color, seed area, seed length, seed width and TGW) that we included in this manuscript. In addition, we have described the SNP calling, GWAS and population genomics analysis in detail in the Methods section.

f. I think that it will be a good idea to contextualize the transposable element expansion for closed related genomes, such as *Vicia sativa* and *Pisum sativum*. This could be done with a violin plot with x-axis associated with TE class and species, and y-axis for TE age (e.g., <https://bmcpantbiol.biomedcentral.com/articles/10.1186/s12870-021-02858-1/figures/3>).

Answer: We would very much have liked to carry out this analysis, but the *Pisum sativum* genome is based on short reads, while the *V. sativa* genome was assembled using ONT data; our *V. faba* assemblies are based on PacBio HiFi data. These differences mean that we cannot guarantee clearly interpretable results in the comparisons and have therefore opted to leave them out. This is especially the case here because the size of the three genomes is connected to the abundance of the *Ogre* retrotransposon; its 20 kb length requires particularly long N50 values for similar rates of incorporation of *Ogre* elements into highly and equally contiguous genomes. See also <https://academic.oup.com/gigascience/article/9/12/giaa123/6034784> for differences between ONT and PacBio assemblies (differences in assembly error rates can affect TE age estimates), point a. on comparisons to short read assemblies and point g. specifically for comparisons with *V. sativa*.

g. Comparison with the *V. sativa* genome is missing. Although the *V. sativa* genome quality may not be as high as this one, I think that the manuscript should have a small paragraph describing the comparison between both genomes.

Answer: During data analysis, we did perform several comparisons with the *V. sativa* genome. However, we ultimately decided not to include them in the manuscript because of quality issues with the *V. sativa* genome assembly. We simultaneously detected synteny between Hedin/2, *V. sativa* and pea using MCScanX and visualized syntenic relationships using Synvisio. While for pea we observed clear syntenic signals between chromosomes, for *V. sativa* the signal was much more noisy and distributed (for example there was no clear correspondence between *V. sativa* chr2 and any of the *V. faba* or pea chromosomes). This suggests underlying issues with *V. sativa* pseudomolecule construction, especially since no genetic map was available for *V. sativa*.

Furthermore, ahead of more detailed gene family analysis (like that for RdDM) we compared overall statistics for high confidence gene models (protein coding gene with intact ORFs annotated as ‘coding’ by CPC2 (Coding Potential Calculator 2) across *V. faba*, *V. sativa*, *P. sativum* and *L. culinaris*. We found that while gene number and median CDS length were very similar for *V. faba*, *P. sativum* and *L. culinaris*, *V. sativa* was a clear outlier, with more, shorter gene models, suggesting fragmentation of annotation.

Species	Version	Full	High confidence	Median filtered CDS size
Lens culinaris	v2.0	58,243	38,732	888
Pisum sativum	v1a	44,756	34,689	885
Vicia sativa *	v1.0	53,218	44,551	825
Vicia faba (Hedin/2)	v1.0	34,221	34,221	906

Vicia faba (Tiffany)	v1.0	34,043	34,043	891
------	--------	--------	-----

h. Discussion section is more of a conclusion than a discussion (it may be a switch in the header of the section).

Answer: We followed the format of the most recent *Nature* paper describing a crop genome (<https://www.nature.com/articles/s41586-022-04732-y>); we are happy to revise this according to the Editor's wishes.

G. References: appropriate credit to previous work?

=====

I think so. I understand that the number of citations may be limited in this manuscript. Nevertheless, it will be useful to read in the introduction that the *Vicia sativa* genome was already published and cite according.

Answer: We have cited the *V. sativa* genome paper in the main text.

H. Clarity and context: lucidity of abstract/summary, appropriateness of abstract, introduction and conclusions

=====

The main text has several part missing that will help with the manuscript reading. Right now, I have the feeling of a lot of work and results compressed into a short manuscript what drive to many questions during its reading. The abstract does not contain information about genome assembly size, number of gene models, and candidate genes associated with the presented trait. The introduction has missing that the *Vicia sativa* genome was published. Probably this type of information could be useful for the reader. I think that the header of the Conclusion section has been mistaken by Discussion (that it is included in the results).

Answer: We have cited the *V. sativa* genome paper in the main text. As mentioned in G., we have followed current *Nature* formatting, but we are happy to edit the abstract and other sections according to editorial requests.

Reviewed by Aureliano Bombarely on October 31st, 2022.

Referee #2 (Remarks to the Author):

Jayakodi and colleagues report a complete genome for the fava bean, an important grain legume. With the support of deep coverage with long reads and Hi-C, the authors can capture nearly 12GB of the approximately 13GB diploid genome. The authors show how transposable elements have expanded rapidly in the lineage leading to fava beans. A complete genome also facilitates an understanding of the evolution of gene families in comparison to other legumes, and patterns of tandem duplication. More important for future breeding, the authors develop dense genomewide markers for trait discovery.

This is a really important paper. Fava bean has immense potential for increasing food and nutritional security and increasing the availability of plant protein as a highly productive cool season legume. Furthermore, this work is a technical achievement, due to the size and complexity of the fava bean genome.

The abundance of high-quality genomic data is presented clearly.

An example of the clear presentation of results is the mapping of hilum color, an important agronomic characteristic. This trait is linked (by 5-10cM) to vicine levels but matters on its own as a quality trait (for reasons that are not well explained, despite the otherwise clear presentation).

Answer: We have added a phrase, at the beginning of the hilum colour mapping section, to explain that human consumers prefer pale hila.

Statistical analyses are appropriate for a genome paper, including assembly, TE evolution, diversity analysis, and GWAS.

I have a handful of suggestions for minor improvement of the manuscript.

Some unique aspects of fava bean are not mentioned until the discussion, such as vicine and favism, or the lack of a compatible wild relative and the challenges this poses for understanding fava bean domestication. This may confuse some readers.

The authors have worked on hilum colour many years. It is not clearly explained by this trait is agronomically significant. It is a great example trait, as the complexity of the repeat cluster harboring the GWA hit would be impossible to resolve without a high quality genome. PPO analysis of metabolites in seeds with varying hilum color needs further explanation.

Fava is unusual due to a lack of compatible wild relative. It is not clear if a wild relative simply has not been found, if the immediate ancestor is extinct, or if sufficient genome evolution occurred in the lineage leading to cultivated fava bean that it became incompatible with other *Vicia* species such as the phenotypically similar *Vicia narbonensis*. This unique aspect of fava is not mentioned until the discussion.

Answer: We agree that these aspects of *V. faba* are well worth mentioning. We have restructured the introduction to focus less on the potential role of faba bean in meeting high demand for plant protein in temperate developed countries (which Reviewer 3 characterized as “overstated”) and more on the mysterious origins of *Vicia faba* and evolution of seed size, the latter point tying in with the novel results on the genetic basis for seed size presented in this revision. We did highlight the recent discovery of the vicine biosynthetic pathway in the introductory section, but stopped short of mentioning favism and would suggest leaving it this way to keep the focus on the crop biology as requested. We have added additional detail on the analysis of hilum colour related metabolites.

As is typical of genome papers at the highest tier journals, the authors have placed far more data and results than can be presented in the space allowed. This is to be expected, but some of the more interesting results are buried in supplemental tables and barely described in the methods. Expanding a few legends will help without great expansion of text. For example, pretzel visualizations and examination of further traits are barely mentioned in the main text. The figure legend for it, and the methods, could allow some description.

Answer: We now added extensive discussion of seed traits in the main body of the manuscript, including new main figure panels (**Figure 3**) and an additional paragraph describing association studies and candidate gene identification. Details of GWAS methods were also added. We have also included a Supplementary Note that details how to log in and use Pretzel to compare the seed size markers identified here with previous QTL studies.

In the same vein, geographic regional differentiation could be better explained. This is something that the supplemental figure and legend might address at more length. The four regions do vary in their patterns of genetic variation. This may give some insight into where fava originated, or how it has shifted with historical migration along with human groups.

Answer: We have updated the **Extended Data Figure 19** legend with regard to the reviewer's concern. We have described the subpopulation clusters related to geographic origin and number of genotypes admixed due to hybridization between subpopulations. Our population is exclusively composed of artificially inbred lines (whatever the origin); we could not include wild genotypes (because the wild progenitor is unknown), and the centre of diversity also remains uncertain. The origin and dispersal routes are therefore difficult to investigate with this diversity panel, which was designed with trait-mapping in mind. A different type of data based on a panel of genotypes including landrace populations and modern elite lines combined with whole-genome re-sequencing at high coverage would be needed to potentially provide insights into the origin and migration of faba bean, by detailed characterisation and analysis of patterns of genetic variation. Such analyses might still be confounded, though, by the transfer and sharing of germplasm suggested by the distribution of seed-enlarging alleles that we present in the revised version.

Referee #3 (Remarks to the Author):

This is an extremely well written manuscript describing the assembly, annotation, and characterization of the faba bean genome. The novelty of the entire manuscript is that this is a huge genome. The authors did a thorough analysis of the genome and overall, it is interesting in how it differs from other large plant genomes that are polyploid or diploid with high repetitive sequence content. I perhaps think this is the most fascinating aspect of the genome that receives limited attention in the manuscript.

Overall the authors performed standard genome analyses to document the quality of their genome (which is high) and how it can be used to identify genes involved in key traits via the example of hilum color in two varieties. The methods are well described and the results clearly shown and interpreted.

The Introduction is highly abbreviated thus a naïve reader will not understand much about faba bean. I do think its positive attributes are overly stated in the Introduction as all of the US and some of lower Canada is soybean production thus I do not believe temperate regions of Europe and America can not yield sufficient plant-based protein. The problem is that the majority of soy is not for human consumption, instead it is animal feed and byproducts.

Answer: We have revised the introduction as suggested.

Colors in Fig 1d difficult to see. Suggest change the maroon and blue with brighter colors to contrast with the dark grey. Explain what the red dotted line is, I assume it is the centromere.

Answer: We have now changed the colours that contrast with the dark grey based on the reviewer's suggestion. In addition, we mentioned the meaning of the red dotted line, which is indeed the centromere, in the figure legend.

The different levels/phases of annotation in Extended data table 3 are confusing. Add a legend to explain these.

Answer: A legend with explanation of terms was added.

Line 236, unclear why Fig 1d is cited.

Answer: Thanks for pointing this out. We have removed it.

More details on the data availability of the genome and annotation is warranted.

Answer: Genome assemblies and corresponding annotations for Hedin/2 and Tiffany are available from <https://projects.au.dk/fabagenome/genomics-data>. A corresponding availability statement specifically referring to genome assemblies and annotation was now added under the 'Data availability' heading along with a genome browser link.

A statement of why Hedin/2 was selected as the initial reference genome is warranted.

Answer: A good point. We have explained this in the results section.

A detailed analysis of gene location vs recombination location would have been very engaging.

Answer: Indeed, it is interesting to check how close recombination breakpoints are to genes. However, SNPs from our SPET data are highly enriched for the gene space, therefore pinpointing actual crossover locations is difficult and will be confounded by the genic SNP locations.

A statement of how (or if) faba bean can be transformed should be added.

Answer: We have added a comment and reference to the Conclusions section

A deeper description of the diversity panel would help learn what genetic bottlenecks there are for faba bean.

Answer: We have updated **Supplementary table 6** with passport descriptions and phenotype data related to our diversity panel. See also our response to the last point of Reviewer 2 for an explanation of the difficulties in addressing genetic bottlenecks using the data type available.

Reviewer Reports on the First Revision:

Referees' comments:

Referee #1 (Remarks to the Author):

I would like to thank the authors for resolving the questions and concerns risen during my first revision. I do not have any other comments and concerns although I will recommend a revision of the small details that may escape this version of the manuscript (e.g., in the material and methods, some of the described tools do not have the version number, for some supplementary tables, the units are not in the header but in the data cell, like in the Extended Table 1...). Congratulations on the nice work presented in your manuscript.

Reviewed by Aureliano Bombarely on January 9th, 2023.

Referee #2 (Remarks to the Author):

The authors have adequately addressed all my concerns from the previous review. I find it improved because of the revisions and the careful attention to reviewer input.

Referee #3 (Remarks to the Author):

The authors did a nice job of answering my previous comments. The additional text, especially in the introduction, will make this manuscript of broader interest. I think to get the reader more excited about faba bean you should include as figure 1a pictures of the leaves, the various seed sizes, and botanical varieties. Some other minor comments are:

Line 63: replace grown with expanded

Line 86: I think to get the reader more excited about faba bean you should include as figure 1a pictures of the leaves, the various seed sizes, and botanical varieties

Line 258: I would point out to the authors just detection of frameshift/premature stops via project of SNPs/indels in non-reference accessions can be misleading as shown by Gan et al. in Arabidopsis (doi:10.1038/nature10414) where most of these 'predicted high impact polymorphisms' did not really exist as in other accessions, an alternative gene model is used.

Line 285: Is it the small-seed population 4 or population 4 which has a small number of members?

Author Rebuttals to First Revision:

Response to reviewer comments

Referee #1 (Remarks to the Author):

I would like to thank the authors for resolving the questions and concerns risen during my first revision. I do not have any other comments and concerns although I will recommend a revision of the small details that may escape this version of the manuscript (e.g., in the material and methods, some of the described tools do not have the version number, for some supplementary tables, the units are not in the header but in the data cell, like in the Extended Table 1...). Congratulations on the nice work presented in your manuscript.

Reviewed by Aureliano Bombarely on January 9th, 2023.

Response:

Thank you for the careful assessment of our manuscript. We have gone through the Methods in detail and added version numbers for all tools where applicable.

Referee #2 (Remarks to the Author):

The authors have adequately addressed all my concerns from the previous review. I find it improved because of the revisions and the careful attention to reviewer input.

Response:

Thank you for the positive comments and for your efforts in helping us to improve the manuscript.

Referee #3 (Remarks to the Author):

The authors did a nice job of answering my previous comments. The additional text, especially in the introduction, will make this manuscript of broader interest. I think to get the reader more excited about faba bean you should include as figure 1a pictures of the leaves, the various seed sizes, and botanical varieties.

Response:

[Redacted image below]

Thank you. Your contribution to evaluating and improving the manuscript is much appreciated.

Some other minor comments are:

Line 63: replace grown with expanded

Response: The suggested change was made.

Line 86: I think to get the reader more excited about faba bean you should include as figure 1a pictures of the leaves, the various seed sizes, and botanical varieties

Response:

It is a good idea to include pictures of faba bean in the figures, but we are already pressed for space. Instead, we have submitted a cover suggestion highlighting the extreme seed size, shape and colour diversity of faba bean, which, if accepted, will surely get readers excited.

Line 258: I would point out to the authors just detection of frameshift/premature stops via project of SNPs/indels in non-reference accessions can be misleading as shown by Gan et al. in Arabidopsis (doi:10.1038/nature10414) where most of these 'predicted high impact polymorphisms' did not really exist as in other accessions, an alternative gene model is used.

The variant effect observed does indeed depend on the gene model. The predictions were based on the current version of annotation. We made predictions only for the 'high quality' gene models with complete coding sequences. The atlas provided is a starting point, but further validation of variant impact will be necessary prior to further functional studies.

Line 285: Is it the small-seed population 4 or population 4 which has a small number of members?

Response: We have rephrased the sentence, so it now reads: "...with the exception of population 4, which comprised relatively few accessions that all harboured the seed-enlarging allele of Vfaba.Hedin2.R1.4g051440"